# Deep Learning on Implicit Neural Representations of Shapes

**Luca De Luigi**\*, **Adriano Cardace**\*, **Riccardo Spezialetti**\*
University of Bologna
{luca.deluigi4, adriano.cardace2@unibo.it, riccardo.spezialetti}@unibo.it

**Pierluigi Zama Ramirez, Samuele Salti, Luigi Di Stefano**
University of Bologna

## Abstract

Implicit Neural Representations (INRs) have emerged in the last few years as a powerful tool to encode continuously a variety of different signals like images, videos, audio and 3D shapes. When applied to 3D shapes, INRs allow to overcome the fragmentation and shortcomings of the popular discrete representations used so far. Yet, considering that INRs consist in neural networks, it is not clear whether and how it may be possible to feed them into deep learning pipelines aimed at solving a downstream task. In this paper, we put forward this research problem and propose inr2vec, a framework that can compute a compact latent representation for an input INR in a single inference pass. We verify that inr2vec can embed effectively the 3D shapes represented by the input INRs and show how the produced embeddings can be fed into deep learning pipelines to solve several tasks by processing exclusively INRs.

## 1 Introduction

Since the early days of computer vision, researchers have been processing images stored as two-dimensional grids of pixels carrying intensity or color measurements. But the world that surrounds us is three dimensional, motivating researchers to try to process also 3D data sensed from surfaces. Unfortunately, representation of 3D surfaces in computers does not enjoy the same uniformity as digital images, with a variety of discrete representations, such as voxel grids, point clouds and meshes, coexisting today. Besides, when it comes to processing by deep neural networks, all these kinds of representations are affected by peculiar shortcomings, requiring complex ad-hoc machinery (Qi et al., 2017b; Wang et al., 2019b; Hu et al., 2022) and/or large memory resources (Maturana & Scherer, 2015). Hence, no standard way to store and process 3D surfaces has yet emerged.

Recently, a new kind of representation has been proposed, which leverages on the possibility of deploying a Multi-Layer Perceptron (MLP) to fit a continuous function that represents *implicitly* a signal of interest (Xie et al., 2021). These representations, usually referred to as Implicit Neural Representations (INRs), have been proven capable of encoding effectively 3D shapes by fitting *signed distance functions (sdf)* (Park et al., 2019; Takikawa et al., 2021; Gropp et al., 2020), *unsigned distance functions (udf)* (Chibane et al., 2020) and *occupancy fields (occ)* (Mescheder et al., 2019; Peng et al., 2020). Encoding a 3D shape with a continuous function parameterized as an MLP decouples the memory cost of the representation from the actual spatial resolution, *i.e.*, a surface with arbitrarily fine resolution can be reconstructed from a fixed number of parameters. Moreover, the same neural network architecture can be used to fit different implicit functions, holding the potential to provide a unified framework to represent 3D shapes.

Due to their effectiveness and potential advantages over traditional representations, INRs are gathering ever-increasing attention from the scientific community, with novel and striking results published more and more frequently (Müller et al., 2022; Martel et al., 2021; Takikawa et al., 2021; Liu et al., 2022). This lead us to conjecture that, in the forthcoming future, INRs might emerge as a standard

---

\*Joint first authorship. We thank also Francesco Ballerini for the results produced during his master thesis.

representation to store and communicate 3D shapes, with repositories hosting digital twins of 3D objects realized only as MLPs becoming commonly available.

An intriguing research question does arise from the above scenario: beyond storage and communication, would it be possible to *process* directly INRs of 3D shapes with deep learning pipelines to solve downstream tasks as it is routinely done today with discrete representations like point clouds or meshes? In other words, would it be possible to process an INR of a 3D shape to solve a downstream task, *e.g.*, shape classification, without reconstructing a discrete representation of the surface?

Since INRs are neural networks, there is no straightforward way to process them. Earlier work in the field, namely OccupancyNetworks (Mescheder et al., 2019) and DeepSDF (Park et al., 2019), fit the whole dataset with a shared network conditioned on a different embedding for each shape. In such formulation, the natural solution to the above mentioned research problem could be to use such embeddings as representations of the shapes in downstream tasks. This is indeed the approach followed by contemporary work (Dupont et al., 2022), which addresses such research problem by using as embedding a latent modulation vector applied to a shared base network. However, representing a whole dataset by a shared network sets forth a difficult learning task, with the network struggling in fitting accurately the totality of the samples (as we show in Appendix A). Conversely, several recent works, like SIREN (Sitzmann et al., 2020b) and others (Sitzmann et al., 2020a; Dupont et al., 2021a; Strümpler et al., 2021; Zhang et al., 2021; Tancik et al., 2020) have shown that, by fitting an individual network to each input sample, one can get high quality reconstructions even when dealing with very complex 3D shapes or images. Moreover, constructing an individual INR for each shape is easier to deploy in the wild, as availability of the whole dataset is not required to fit an individual shape. Such works are gaining ever-increasing popularity and we are led to believe that fitting an individual network is likely to become the common practice in learning INRs.

Thus, in this paper, we investigate how to perform downstream tasks with deep learning pipelines on shapes represented as individual INRs. However, a single INR can easily count hundreds of thousands of parameters, though it is well known that the weights of a deep model provide a vastly redundant parametrization of the underlying function (Frankle & Carbin, 2018; Choudhary et al., 2020). Hence, we settle on investigating whether and how an answer to the above research question may be provided by a representation learning framework that learns to squeeze individual INRs into compact and meaningful embeddings amenable to pursuing a variety of downstream tasks.

Our framework, dubbed `inr2vec` and shown in Fig. 1, has at its core an encoder designed to produce a task-agnostic embedding representing the input INR by processing only the INR weights. These embeddings can be seamlessly used in downstream deep learning pipelines, as we validate experimentally for a variety of tasks, like classification, retrieval, part segmentation, unconditioned generation, surface reconstruction and completion. Interestingly, since embeddings obtained from INRs live in low-dimensional vector spaces regardless of the underlying implicit function, the last two tasks can be solved by learning a simple mapping between the embeddings produced with our framework, *e.g.*, by transforming the INR of a *udf* into the INR of an *sdf*. Moreover, `inr2vec` can learn a smooth latent space conducive to interpolating INRs representing unseen 3D objects. Additional details and code can be found at `https://cvlab-unibo.github.io/inr2vec`. Our contributions can be summarised as follows:

- we propose and investigate the novel research problem of applying deep learning directly on individual INRs representing 3D shapes;
- to address the above problem, we introduce `inr2vec`, a framework that can be used to obtain a meaningful compact representation of an input INR by processing only its weights, without sampling the underlying implicit function;
- we show that a variety of tasks, usually addressed with representation-specific and complex frameworks, can indeed be performed by deploying simple deep learning machinery on INRs embedded by `inr2vec`, the same machinery regardless of the INRs underlying signal.

## 2 RELATED WORK

**Deep learning on 3D shapes.** Due to their regular structure, voxel grids have always been appealing representations for 3D shapes and several works proposed to use 3D convolutions to perform both discriminative (Maturana & Scherer, 2015; Qi et al., 2016; Song & Xiao, 2016) and generative tasks

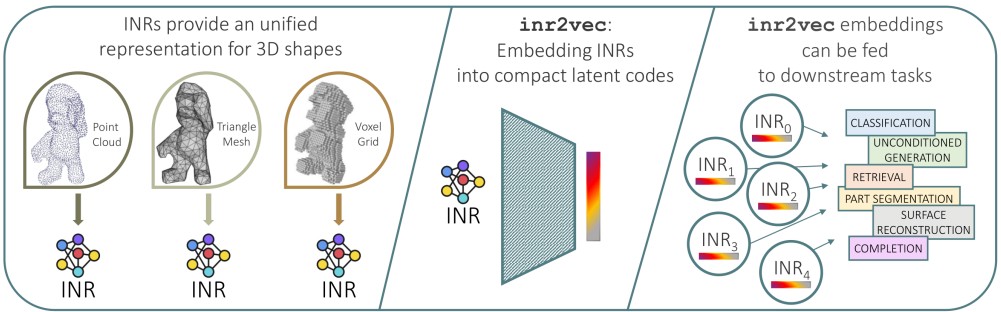

Figure 1: **Overview of our framework. Left**: INRs hold the potential to provide an unified representation for 3D shapes. **Center**: Our framework, dubbed `inr2vec`, produces a compact representation for an input INR by looking only at its weights. **Right**: `inr2vec` embeddings can be used with standard deep learning machinery to solve a variety of downstream tasks.

(Choy et al., 2016; Girdhar et al., 2016; Jimenez Rezende et al., 2016; Stutz & Geiger, 2018; Wu et al., 2016; 2015). The huge memory requirements of voxel-based representations, though, led researchers to look for less demanding alternatives, such as point clouds. Processing point clouds, however, is far from straightforward because of their unorganized nature. As a possible solution, some works projected the original point clouds to intermediate regular grid structures such as voxels (Shi et al., 2020) or images (You et al., 2018; Li et al., 2020a). Alternatively, PointNet (Qi et al., 2017a) proposed to operate directly on raw coordinates by means of shared multi-layer perceptrons followed by max pooling to aggregate point features. PointNet++ (Qi et al., 2017b) extended PointNet with a hierarchical feature learning paradigm to capture the local geometric structures. Following PointNet++, many works focused on designing new local aggregation operators (Liu et al., 2020), resulting in a wide spectrum of specialized methods based on convolution (Hua et al., 2018; Xu et al., 2018; Atzmon et al., 2018; Wu et al., 2019; Fan et al., 2021; Xu et al., 2021; Thomas et al., 2019), graph (Li et al., 2019; Wang et al., 2019a;b), and attention (Guo et al., 2021; Zhao et al., 2021) operators. Yet another completely unrelated set of deep learning methods have been developed to process surfaces represented as meshes, which differ in the way they exploit vertices, edges and faces as input data (Hu et al., 2022). *Vertex-based* approaches leverage the availability of a regular domain to encode the knowledge about points neighborhoods through convolution or kernel functions (Masci et al., 2015; Boscaini et al., 2016; Huang et al., 2019; Yang et al., 2020; Haim et al., 2019; Schult et al., 2020; Monti et al., 2017; Smirnov & Solomon, 2021). *Edge-based* methods take advantages of these connections to define an ordering invariant convolution (Hanocka et al., 2019), to construct a graph on the input meshes (Milano et al., 2020) or to navigate the shape structure (Lahav & Tal, 2020). Finally, *Face-based* works extract information from neighboring faces (Lian et al., 2019; Feng et al., 2019; Li et al., 2020b; Hertz et al., 2020). In this work, we explore INRs as a unified representation for 3D shapes and propose a framework that enables the use of the same standard deep learning machinery to process them, independently of the INRs underlying signal.

**3D Implicit neural representations (INRs).** Recent approaches have shown the ability of MLPs to represent implicitly diverse kinds of data such as objects (Genova et al., 2019; 2020; Park et al., 2019; Atzmon & Lipman, 2020; Michalkiewicz et al., 2019; Gropp et al., 2020) and scenes (Sitzmann et al., 2019; Jiang et al., 2020; Peng et al., 2020; Chabra et al., 2020). The works focused on representing 3D data by MLPs rely on fitting implicit functions such as unsigned distance (for point clouds), signed distance (for meshes) (Atzmon & Lipman, 2020; Park et al., 2019; Gropp et al., 2020; Sitzmann et al., 2019; Jiang et al., 2020; Peng et al., 2020) or occupancy (for voxel grids) (Mescheder et al., 2019; Chen & Zhang, 2019). In addition to representing shapes, some of these models have been extended to encode also object appearance (Saito et al., 2019; Mildenhall et al., 2020; Sitzmann et al., 2019; Oechsle et al., 2019; Niemeyer et al., 2020), or to include temporal information (Niemeyer et al., 2019). Among these approaches, sinusoidal representation networks (SIREN) (Sitzmann et al., 2020b) use periodical activation functions to capture the high frequency details of the input data. In our work, we focus on implicit neural representations of 3D data represented as voxel grids, meshes and point clouds and we use SIREN as our main architecture.

**Deep learning on neural networks.** Few works attempted to process neural networks by means of other neural networks. For instance, (Unterthiner et al., 2020) takes as input the weights of a network

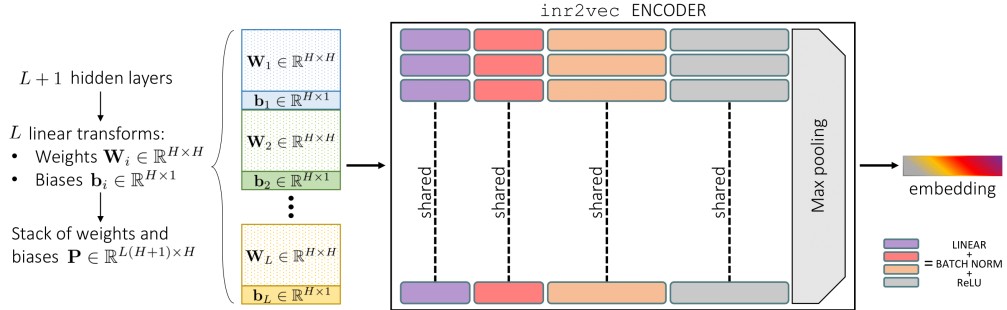

Figure 2: `inr2vec` **encoder architecture**.

and predicts its classification accuracy. (Schürholt et al., 2021) learns a network representation with a self-supervised learning strategy applied on the $N$-dimensional weight array, and then uses the learned representations to predict various characteristics of the input classifier. (Knyazev et al., 2021; Jaeckle & Kumar, 2021; Lu & Kumar, 2020) represent neural networks as computational graphs, which are processed by a GNN to predict optimal parameters, adversarial examples, or branching strategies for neural network verification. All these works see neural networks as algorithms and focus on predicting properties such as their accuracy. On the contrary, we process networks that represent implicitly 3D shapes to perform a variety of tasks directly from their weights, *i.e.*, we treat neural networks as input/output data. To the best of our knowledge, processing 3D shapes represented as INRs has been attempted only in contemporary work (Dupont et al., 2022). However, they rely on a shared network conditioned on shape-specific embeddings while we process the weights of individual INRs, that better capture the underlying signal and are easier to deploy in the wild.

## 3 LEARNING TO REPRESENT INRS

The research question we address in this paper is whether and how can we process directly INRs to perform downstream tasks. For instance, can we classify a 3D shape that is implicitly encoded in an INR? And how? As anticipated in Section 1, we propose to rely on a representation learning framework to squeeze the redundant information contained in the weights of INRs into compact latent codes that could be conveniently processed with standard deep learning pipelines.

Our framework, dubbed `inr2vec`, is composed of an encoder and a decoder. The encoder, detailed in Fig. 2, is designed to take as input the weights of an INR and produce a compact embedding that encodes all the relevant information of the input INR. A first challenge in designing an encoder for INRs consists in defining how the encoder should ingest the weights as input, since processing naively all the weights would require a huge amount of memory (see Appendix E). Following standard practice (Sitzmann et al., 2020b;a; Dupont et al., 2021a; Strümpler et al., 2021; Zhang et al., 2021), we consider INRs composed of several hidden layers, each one with $H$ nodes, *i.e.*, the linear transformation between two consecutive layers is parameterized by a matrix of weights $\mathbf{W}_i \in \mathbb{R}^{H \times H}$ and a vector of biases $\mathbf{b}_i \in \mathbb{R}^{H \times 1}$. Thus, stacking $\mathbf{W}_i$ and $\mathbf{b}_i^T$, the mapping between two consecutive layers can be represented by a single matrix $\mathbf{P}_i \in \mathbb{R}^{(H+1) \times H}$. For an INR composed of $L + 1$ hidden layers, we consider the $L$ linear transformations between them. Hence, stacking all the $L$ matrices $\mathbf{P}_i \in \mathbb{R}^{(H+1) \times H}, i = 1, \dots, L$, between the hidden layers we obtain a single matrix $\mathbf{P} \in \mathbb{R}^{L(H+1) \times H}$, that we use to represent the INR in input to `inr2vec` encoder. We discard the input and output layers in our formulation as they feature different dimensionality and their use does not change `inr2vec` performance, as shown in Appendix I.

The `inr2vec` encoder is designed with a simple architecture, consisting of a series of linear layers with batch norm and ReLU non-linearity followed by final max pooling. At each stage, the input matrix is transformed by one linear layer, that applies the same weights to each row of the matrix. The final max pooling compresses all the rows into a single one, obtaining the desired embedding. It is worth observing that the randomness involved in fitting an individual INR (weights initialization, data shuffling, etc.) causes the weights in the same position in the INR architecture not to share the same role across INRs. Thus, `inr2vec` encoder would have to deal with input vectors whose elements capture different information across the different data samples, making it impossible to train

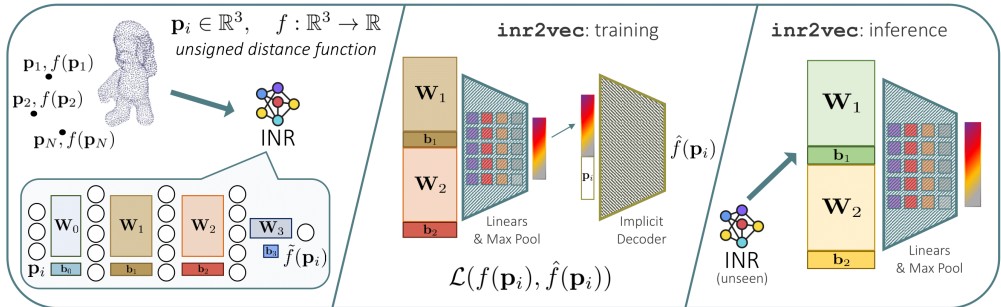

Figure 3: **Training and inference of our framework. Left:** We consider shapes represented as INRs. As an example, we show an INR fitting the *udf* of a surface. **Center:** `inr2vec` encoder is trained together with an implicit decoder to replicate the underlying 3D signal of the input INR. **Right:** At inference time, the learned encoder can be used to obtain a compact embedding from unseen INRs.

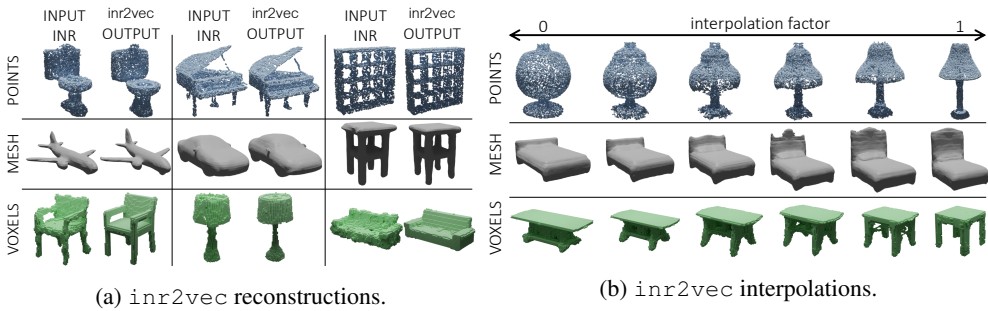

(a) `inr2vec` reconstructions.          (b) `inr2vec` interpolations.

Figure 4: **Properties of `inr2vec` latent space.**

the framework. However, the use of a shared, pre-computed initialization has been advocated as a good practice when fitting INRs, *e.g.*, to reduce training time by means of meta-learned initialization vectors, as done in MetaSDF (Sitzmann et al., 2020a) and in the contemporary work exploring processing of INRs (Dupont et al., 2022), or to obtain desirable geometric properties (Gropp et al., 2020). We empirically found that following such a practice, *i.e.*, initializing all INRs with the same random vector, favours alignment of weights across INRs and enables convergence of our framework.

In order to guide the `inr2vec` encoder to produce meaningful embeddings, we first note that we are not interested in encoding the values of the input weights in the embeddings produced by our framework, but, rather, in storing information about the 3D shape represented by the input INR. For this reason, we supervise the decoder to replicate the function approximated by the input INR instead of directly reproducing its weights, as it would be the case in a standard auto-encoder formulation. In particular, during training, we adopt an implicit decoder inspired by (Park et al., 2019), which takes in input the embeddings produced by the encoder and decodes the input INRs from them (see Fig. 3 center). More specifically, when the `inr2vec` encoder processes a given INR, we use the underlying signal to create a set of 3D queries $\mathbf{p}_i$, paired with the values $f(\mathbf{p}_i)$ of the function approximated by the input INR at those locations (the type of function depends on the underlying signal modality, it can be *udf* in case of point clouds, *sdf* in case of triangle meshes or *occ* in case of voxel grids). The decoder takes in input the embedding produced by the encoder concatenated with the 3D coordinates of a query $\mathbf{p}_i$ and the whole encoder-decoder is supervised to regress the value $f(\mathbf{p}_i)$. After the overall framework has been trained end to end, the frozen encoder can be used to compute embeddings of unseen INRs with a single forward pass (see Fig. 3 right) while the implicit decoder can be used, if needed, to reconstruct the discrete representation given an embedding.

In Fig. 4a we compare 3D shapes reconstructed from INRs unseen during training with those reconstructed by the `inr2vec` decoder starting from the latent codes yielded by the encoder. We visualize point clouds with 8192 points, meshes reconstructed by marching cubes (Lorensen & Cline, 1987) from a grid with resolution $128^3$ and voxels with resolution $64^3$. We note that, though our embedding is dramatically more compact than the original INR, the reconstructed shape resembles the ground-truth with a good level of detail. Moreover, in Fig. 4b we linearly interpolate between the

| Method | ModelNet40 | | | ShapeNet10 | | | ScanNet10 | | |
|---|---|---|---|---|---|---|---|---|---|
| | mAP@1 | mAP@5 | mAP@10 | mAP@1 | mAP@5 | mAP@10 | mAP@1 | mAP@5 | mAP@10 |
| PointNet (Qi et al., 2017a) | 80.1 | 91.7 | 94.4 | 90.6 | 96.6 | 98.1 | 65.7 | 86.2 | 92.6 |
| PointNet++ (Qi et al., 2017b) | 85.1 | 93.9 | 96.0 | 92.2 | 97.5 | 98.6 | 71.6 | 89.3 | 93.7 |
| DGCNN (Wang et al., 2019b) | 83.2 | 92.7 | 95.1 | 91.0 | 96.7 | 98.2 | 66.1 | 88.0 | 93.1 |
| `inr2vec` | 81.7 | 92.6 | 95.1 | 90.6 | 96.7 | 98.1 | 65.2 | 87.5 | 94.0 |

Table 1: **Point cloud retrieval quantitative results.**

| Method | Point Cloud | | | Mesh | Voxels |
|---|---|---|---|---|---|
| | ModelNet40 | ShapeNet10 | ScanNet10 | Manifold40 | ShapeNet10 |
| PointNet (Qi et al., 2017a) | 88.8 | 94.3 | 72.7 | – | – |
| PointNet++ (Qi et al., 2017b) | 89.7 | 94.6 | 76.4 | – | – |
| DGCNN (Wang et al., 2019b) | 89.9 | 94.3 | 76.2 | – | – |
| MeshWalker (Lahav & Tal, 2020) | – | – | – | 90.0 | – |
| Conv3DNet (Maturana & Scherer, 2015) | – | – | – | – | 92.1 |
| `inr2vec` | 87.0 | 93.3 | 72.1 | 86.3 | 93.0 |

Table 2: **Results on shape classification across representations.**

embeddings produced by `inr2vec` from two input shapes and show the shapes reconstructed from the interpolated embeddings. Results highlight that the latent space learned by `inr2vec` enables smooth interpolations between shapes represented as INRs. *Additional details on `inr2vec` training and the procedure to reconstruct the discrete representations from the decoder are in the Appendices.*

## 4 DEEP LEARNING ON INRS

In this section, we first present the set-up of our experiments. Then, we show how several tasks dealing with 3D shapes can be tackled by working only with `inr2vec` embeddings as input and/or output. *Additional details on the architectures and on the experimental settings are in Appendix F.*

**General settings.** In all the experiments reported in this section, we convert 3D discrete representations into INRs featuring 4 hidden layers with 512 nodes each, using the SIREN activation function (Sitzmann et al., 2020b). We train `inr2vec` using an encoder composed of four linear layers with respectively 512, 512, 1024 and 1024 features, embeddings with 1024 values and an implicit decoder with 5 hidden layers with 512 features. In all the experiments, the baselines are trained using standard data augmentation (random scaling and point-wise jittering), while we train both `inr2vec` and the downstream task-specific networks on datasets augmented offline with the same transformations.

**Point cloud retrieval.** We first evaluate the feasibility of using `inr2vec` embeddings of INRs to solve tasks usually tackled by representation learning, and we select 3D retrieval as a benchmark. We follow the procedure introduced in (Chang et al., 2015), using the euclidean distance to measure the similarity between embeddings of unseen point clouds from the test sets of ModelNet40 (Wu et al., 2015) and ShapeNet10 (a subset of 10 classes of the popular ShapeNet dataset (Chang et al., 2015)). For each embedded shape, we select its $k$-nearest-neighbours and compute a Precision Score comparing the classes of the query and the retrieved shapes, reporting the mean Average Precision for different $k$ (mAP@$k$). Beside `inr2vec`, we consider three baselines to embed point clouds, which are obtained by training the PointNet (Qi et al., 2017a), PointNet++ (Qi et al., 2017b) and DGCNN (Wang et al., 2019b) encoders in combination with a fully connected decoder similar to that proposed in (Fan et al., 2017) to reconstruct the input cloud. Quantitative results, reported in Table 1, show that, while there is an average gap of 1.8 mAP with PointNet++, `inr2vec` is able to match, and in some cases even surpass, the performance of the other baselines. Moreover, it is possible to appreciate in Fig. 5 that the retrieved shapes not only belong to the same class as the query but present also the same coarse structure. This finding highlights how the pretext task used to learn `inr2vec` embeddings can summarise relevant shape information effectively.

**Shape classification.** We then address the problem of classifying point clouds, meshes and voxel grids. For point clouds we use three datasets: ShapeNet10, ModelNet40 and ScanNet10 (Dai et al., 2017). When dealing with meshes, we conduct our experiments on the Manifold40 dataset (Hu et al., 2022). Finally, for voxel grids, we use again ShapeNet10, quantizing clouds to grids with resolution $64^3$. Despite the different nature of the discrete representations taken into account, `inr2vec` allows us to perform shape classification on INRs embeddings, augmented online with E-Stitchup (Wolfe & Lundgaard, 2019), by the very same downstream network architecture, *i.e.*, a simple fully connected classifier consisting of three layers with 1024, 512 and 128 features. We consider as baselines well-known architectures that are optimized to work on the specific input representations of each

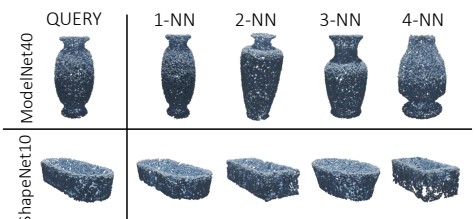

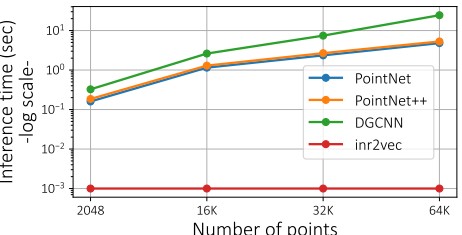

Figure 5: **Point cloud retrieval qualitative results.** Given the `inr2vec` embedding of a query shape, we show the shapes reconstructed from the closest embeddings (L2 distance).

Figure 6: **Time required to classify INRs encoding udf.** For point cloud classifiers, the time to reconstruct the discrete point cloud from the INR is included in the chart.

| Method | instance mIoU | class mIoU | airplane | bag | cap | car | chair | earphone | guitar | knife | lamp | laptop | motor | mug | pistol | rocket | skateboard | table |
|---|---|---|---|---|---|---|---|---|---|---|---|---|---|---|---|---|---|---|
| PointNet (Qi et al., 2017a) | 83.1 | 78.96 | 81.3 | 76.9 | 79.6 | 71.4 | 89.4 | 67.0 | 91.2 | 80.5 | 80.0 | 95.1 | 66.3 | 91.3 | 80.6 | 57.8 | 73.6 | 81.5 |
| PointNet++ (Qi et al., 2017b) | 84.9 | 82.73 | 82.2 | 88.8 | 84.0 | 76.0 | 90.4 | 80.6 | 91.8 | 84.9 | 84.4 | 94.9 | 72.2 | 94.7 | 81.3 | 61.1 | 74.1 | 82.3 |
| DGCNN (Wang et al., 2019b) | 83.6 | 80.86 | 80.7 | 84.3 | 82.8 | 74.8 | 89.0 | 81.2 | 90.1 | 86.4 | 84.0 | 95.4 | 59.3 | 92.8 | 77.8 | 62.5 | 71.6 | 81.1 |
| inr2vec | 81.3 | 76.91 | 80.2 | 76.2 | 70.3 | 70.1 | 88.0 | 65.0 | 90.6 | 82.1 | 77.4 | 94.4 | 61.4 | 92.7 | 79.0 | 56.2 | 68.6 | 78.5 |

Table 3: **Part segmentation quantitative results.** We report the IoU for each class, the mean IoU over all the classes (class mIoU) and the mean IoU over all the instances (instance mIoU).

dataset. For point clouds, we consider PointNet (Qi et al., 2017a), PointNet++ (Qi et al., 2017b) and DGCNN (Wang et al., 2019b). For meshes, we consider a recent and competitive baseline that processes directly triangle meshes (Lahav & Tal, 2020). As for voxel grids, we train a 3D CNN classifier that we implemented following (Maturana & Scherer, 2015) (Conv3DNet from now on). Since only the train and test splits are released for all the datasets, we created validation splits from the training sets in order to follow a proper train/val protocol for both the baselines and `inr2vec`. As for the test shapes, we evaluated all the baselines on the discrete representations reconstructed from the INRs fitted on the original test sets, as these would be the only data available at test time in a scenario where INRs are used to store and communicate 3D data. We report the results of the baselines tested on the original discrete representations available in the original datasets in Appendix H: they are in line with those provided here. The results in Table 2 show that `inr2vec` embeddings deliver classification accuracy close to the specialized baselines across all the considered datasets, regardless of the original discrete representation of the shapes in each dataset. Remarkably, our framework allows us to apply the same simple classification architecture on all the considered input modalities, in stark contrast with all the baselines that are highly specialized for each modality, exploit inductive biases specific to each such modality and cannot be deployed on representations different from those they were designed for. Furthermore, while presenting a gap of some accuracy points *w.r.t.* the most recent architectures, like DGCNN and MeshWalker, the simple fully connected classifier that we applied on `inr2vec` embeddings obtains scores comparable to standard baselines like PointNet and Conv3DNet. We also highlight that, should 3D shapes be stored as INRs, classifying them with the considered specialized baselines would require recovering the original discrete representations by the lengthy procedures described in Appendix C. Thus, in Fig. 6, we report the inference time of standard point cloud classification networks while including also the time needed to reconstruct the discrete point cloud from the input INR of the underlying *udf* at different resolutions. Even at the coarsest resolution (2048 points), all the baselines yield an inference time which is one order of magnitude higher than that required to classify directly the `inr2vec` embeddings. Increasing the resolution of the reconstructed clouds makes the inference time of the baselines prohibitive, while `inr2vec`, not requiring the explicit clouds, delivers a constant inference time of 0.001 seconds.

**Point cloud part segmentation.** While the tasks of classification and retrieval concern the possibility of using `inr2vec` embeddings as a compact proxy for the global information of the input shapes, with the task of point cloud part segmentation we aim at investigating whether `inr2vec` embeddings can be used also to assess upon local properties of shapes. The part segmentation task consists in predicting a semantic (*i.e.*, part) label for each point of a given cloud. We tackle this problem by training a decoder similar to that used to train our framework (see Fig. 7a). Such decoder is fed with the `inr2vec` embedding of the INR representing the input cloud, concatenated with the coordinate

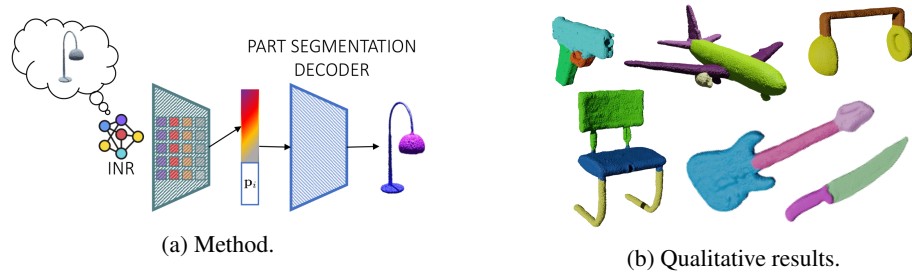

(a) Method.

(b) Qualitative results.

Figure 7: **Point cloud part segmentation.**

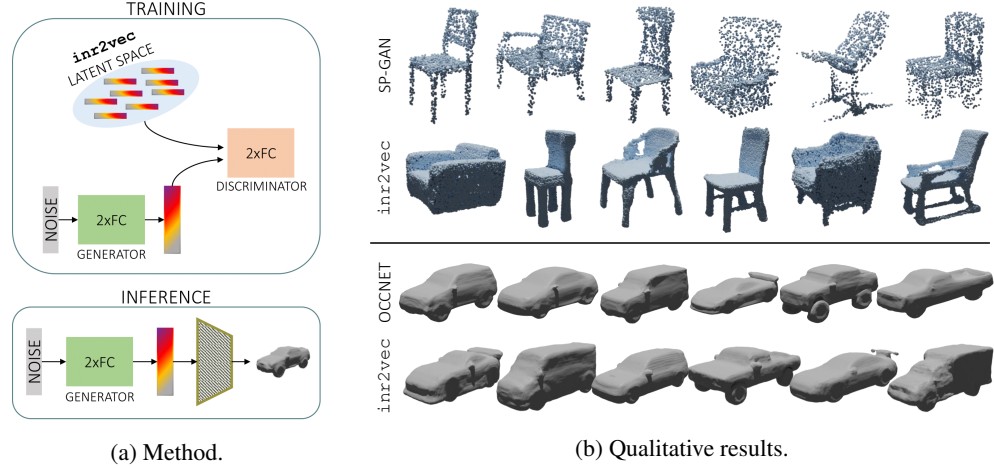

(a) Method.

(b) Qualitative results.

Figure 8: **Learning to generate shapes from** `inr2vec` **latent space.**

of a 3D query, and it is trained to predict the label of the query point. We train it, as well as PointNet, PointNet++ and DGCNN, on the ShapeNet Part Segmentation dataset (Yi et al., 2016) with point clouds of 2048 points, with the same train/val/test of the classification task. Quantitative results reported in Table 3 show the possibility of performing also a local discriminative task as challenging as part segmentation based on the task-agnostic embeddings produced by `inr2vec` and, in so doing, to reach performance not far from that of specialized architectures. Additionally, in Fig. 7b we show point clouds reconstructed at 100K points from the input INRs and segmented with high precision thanks to our formulation based on a semantic decoder conditioned by the `inr2vec` embedding.

**Shape generation.** With the experiments reported above we validated that, thanks to `inr2vec` embeddings, INRs can be used as input in standard deep learning machinery. In this section, we address instead the task of shape generation in an adversarial setting to investigate whether the compact representations produced by our framework can be adopted also as medium for the output of deep learning pipelines. For this purpose, as depicted in Fig. 8a, we train a Latent-GAN (Achlioptas et al., 2018) to generate embeddings indistinguishable from those produced by `inr2vec` starting from random noise. The generated embeddings can then be decoded into discrete representations with the implicit decoder exploited during `inr2vec` training. Since our framework is agnostic *w.r.t.* the original discrete representation of shapes used to fit INRs, we can train Latent-GANs with embeddings representing point clouds or meshes based on the same identical protocol and architecture (two simple fully connected networks as generator and discriminator). For point clouds, we train one Latent-GAN on each class of ShapeNet10, while we use models of cars provided by (Mescheder et al., 2019) when dealing with meshes. In Fig. 8b, we show some samples generated with the described procedure, comparing them with SP-GAN (Li et al., 2021) on the *chair* class for what concerns point clouds and Occupancy Networks (Mescheder et al., 2019) (VAE formulation) for meshes. Generated examples of other classes for point clouds are shown in Appendix J. The shapes generated with our Latent-GAN trained only on `inr2vec` embeddings seem comparable to those produced by the considered baselines, in terms of both diversity and richness of details. Additionally, by generating embeddings that represent implicit functions, our method enables sampling point

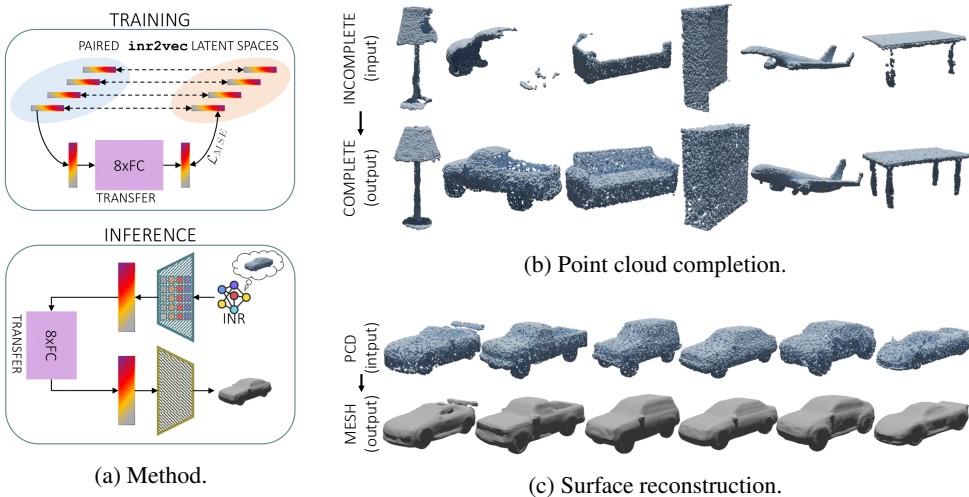

(a) Method.

(b) Point cloud completion.

(c) Surface reconstruction.

Figure 9: **Learning a mapping between** `inr2vec` **latent spaces.**

clouds at any arbitrary resolution (*e.g.*, 8192 points in Fig. 8b) whilst SP-GAN would require a new training for each desired resolution since the number of generated points must be set at training time.

**Learning a mapping between** `inr2vec` **embedding spaces.** We have shown that `inr2vec` embeddings can be used as a proxy to feed INRs in input to deep learning pipelines, and that they can also be obtained as output of generative frameworks. In this section we move a step further, considering the possibility of learning a mapping between two distinct latent spaces produced by our framework for two separate datasets of INRs, based on a *transfer* function designed to operate on `inr2vec` embeddings as both input and output data. Such transfer function can be realized by a simple fully connected network that maps the input embedding into the output one and is trained by a standard MSE loss (see Fig. 9a). As `inr2vec` generates compact embeddings of the same dimension regardless of the input INR modality, the transfer function described here can be applied seamlessly to a great variety of tasks, usually tackled with ad-hoc frameworks tailored to specific input/output modalities. Here, we apply this idea to two tasks. Firstly, we address point cloud completion on the dataset presented in (Pan et al., 2021) by learning a mapping from `inr2vec` embeddings of INRs that represent incomplete clouds to embeddings associated with complete clouds. Then, we tackle the task of surface reconstruction on ShapeNet cars, training the transfer function to map `inr2vec` embeddings representing point clouds into embeddings that can be decoded into meshes. As we can appreciate from the samples in Fig. 9b and Fig. 9c, the transfer function can learn an effective mapping between `inr2vec` latent spaces. Indeed, by processing exclusively INRs embedding, we can obtain output shapes that are highly compatible with the input ones whilst preserving the distinctive details, like the pointy wing of the airplane in Fig. 9b or the flap of the first car in Fig. 9c.

## 5 CONCLUDING REMARKS

We have shown that it is possible to apply deep learning directly on individual INRs representing 3D shapes. Our formulation of this novel research problem leverages on a task-agnostic encoder which embeds INRs into compact and meaningful latent codes without accessing the underlying implicit function. Our framework ingests INRs obtained from different 3D discrete representations and performs various tasks through standard machinery. However, we point out two main limitations: i) Although INRs capture continuous geometric cues, `inr2vec` embeddings achieve results inferior to state-of-the-art solutions ii) There is no obvious way to perform online data augmentation on shapes represented as INRs by directly altering their weights. In the future, we plan to investigate these shortcomings as well as applying `inr2vec` to other input modalities like images, audio or radiance fields. We will also investigate weight-space symmetries (Entezari et al., 2021) as a different path to favour alignment of weights across INRs despite the randomness of training. We reckon that our work may foster the adoption of INRs as a unified and standardized 3D representation, thereby overcoming the current fragmentation of discrete 3D data and associated processing architectures.

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

# A INDIVIDUAL INRS VS. SHARED NETWORK FRAMEWORKS

In this section we aim at comparing the representation power of individual INRs (*i.e.*, one network for each data point) with the one of frameworks adopting a single shared network for the whole dataset, like DeepSDF (Park et al., 2019), OccupancyNetworks (Mescheder et al., 2019) or (Dupont et al., 2022). The important difference between such approaches and our method relies in the fact that in shared network frameworks, the shared network and the set of latent codes are the implicit representation, whose reconstruction quality is negatively affected by using a single network to represent the whole dataset, as shown below. In our framework, instead, we decouple the representations (INRs) from the embeddings used to process them in downstream tasks (yielded by inr2vec). The quality of the representation is then entrusted to the individual INRs and inr2vec does not influence it.

To compare the representation quality of individual INRs with the one of share network frameworks, we fitted the SDF of the meshes in the Manifold40 dataset with OccupancyNetworks (Mescheder et al., 2019), DeepSDF (Park et al., 2019) and LatentModulatedSiren (*i.e.*, the architecture used by the contemporary work that addresses deep learning on INRs (Dupont et al., 2022)). Then, we reconstructed the training discrete meshes from the three frameworks and we compared them with the ground-truth ones, performing the same comparison using the discrete shapes reconstructed from individual INRs. To perform the comparison, we first reconstructed meshes and then we sampled dense point clouds (16,384 points) from the reconstructed surfaces, doing the same for the ground-truth meshes. We report the quantitative comparisons in Table 4, using two metrics: the Chamfer Distance as defined in (Fan et al., 2017) and the F-Score as defined in (Tatarchenko et al., 2019).

| Method | train set | | Method | test set | |
|---|---|---|---|---|---|
| | CD (mm) ↓ | F-Score ↑ | | CD (mm) ↓ | F-Score ↑ |
| OccupancyNetworks | 0.8 | 0.44 | OccupancyNetworks | 1.3 | 0.39 |
| DeepSDF | 11.1 | 0.14 | DeepSDF | 6.6 | 0.25 |
| LatentModulatedSiren | 0.7 | 0.37 | LatentModulatedSiren | 1.9 | 0.28 |
| Individual INRs | **0.3** | **0.50** | Individual INRs | **0.3** | **0.49** |
| (a) Train set. | | | (b) Test set. | | |

Table 4: **Individual INRs vs. shared network frameworks.** Comparison between discrete meshes reconstructed from individual INRs and from shared network frameworks on Manifold40.

The comparison reported in the Table 4a shows that both OccupancyNetworks and LatentModulated-Siren cannot represent the shapes of the training set with a good fidelity, most likely because of the single shared network that struggles to fit a big number of shapes with high variability (∼10K shapes, 40 classes). At the same time, DeepSDF obtains really poor scores, highlighting the difficulty of training an auto-decoder framework on a large and varied dataset. Individual INRs, instead, produce reconstructions with good quality, even if we adopted a fitting procedure with only 500 steps for each shape.

Moreover, the approaches based on a conditioned shared network tend to fail in representing unseen samples that are out of the training distribution. Hence, in the foreseen scenario where INRs become a standard representation for 3D data hosted in public repositories, leveraging on a single shared network may imply the need to frequently retrain the model upon uploading new samples which, in turn, would change the embeddings of all the previously stored data. On the contrary, uploading the repository with a new shape would not cause any sort of issue with individual INRs, where one learns a network for each data point.

To better support our statements, in Table 4b we report the comparison between shape reconstructed from OccupancyNetworks, DeepSDF, LatentModulatedSiren and individual INRs, using shapes from the test set of Manifold40, *i.e.*, shapes unseen at training time. The numbers show that both OccupancyNetworks and LatentModulatedSiren present a drop in the quality of the reconstructions, indicating that both frameworks struggle to represent new shapes not available at training time. Surprisingly, DeepSDF produces better scores on the test set *w.r.t.* the results on the train set, but still presenting a quite poor performance in comparison with the other methods. Conversely, the problem of representing unseen shapes is inherently inexistent when adopting individual INRs, as shown by the numbers in the last row of Table 4b, which are almost identical to the ones presented in Table 4a.

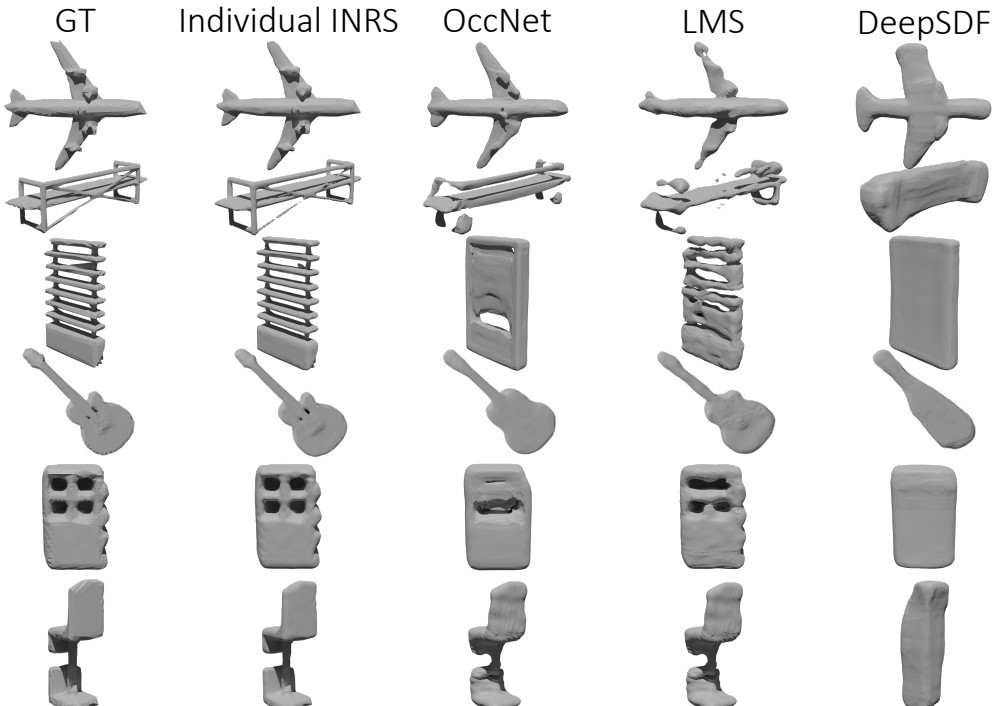

Figure 10: **Individual INRs vs. shared network frameworks (train shapes).** Qualitative comparison of meshes from Manifold40 reconstructed from individual INRs or from shared network frameworks, when dealing with training shapes. OccNet stands for OccupancyNetworks, LMS stands for LatentModulatedSiren.

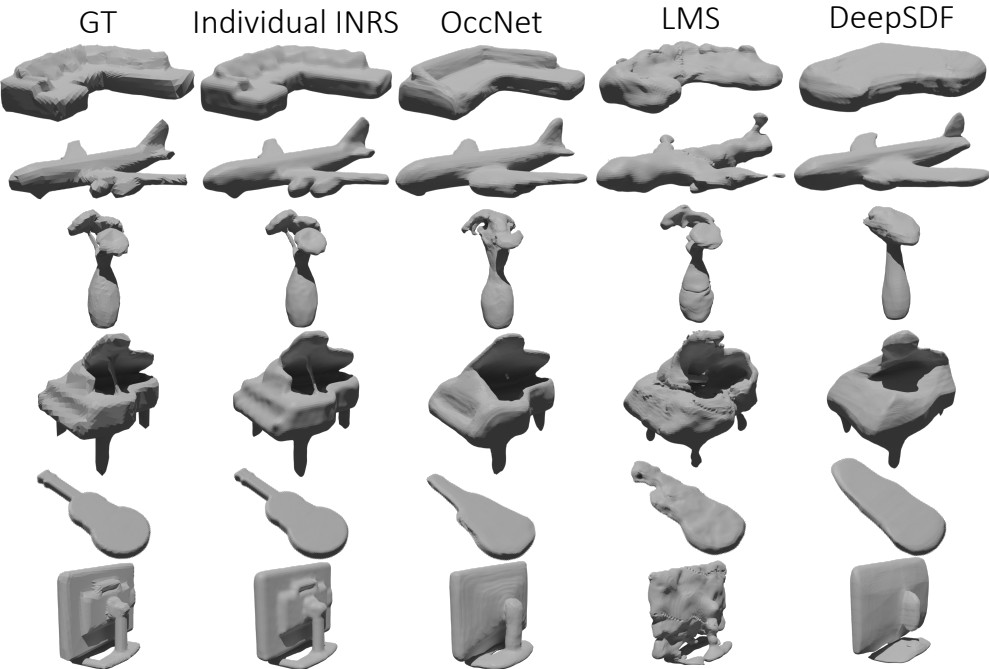

Figure 11: **Individual INRs vs. shared network frameworks (test shapes).** Qualitative comparison of meshes from Manifold40 reconstructed from individual INRs or from shared network frameworks, when dealing with shapes unseed during training. OccNet stands for OccupancyNetworks, LMS stands for LatentModulatedSiren.

We report in Fig. 10 and Fig. 11 the comparison described above from a qualitative perspective. It is possible to observe that the visualizations confirm the results posted in Table 4, with shared network frameworks struggling to represent properly the ground-truth shapes, while individual INRs enable high fidelity reconstructions.

We believe that these results highlight that frameworks based on a single shared network cannot be used as medium to represent shapes as INRs, because of their limited representation power when dealing with large and varied datasets and because of their difficulty in representing new shapes not available at training time.

## B    OBTAINING INRs FROM 3D DISCRETE REPRESENTATIONS

In this section, we detail the procedure used when fitting INRs to create the datasets used in this work. Given a dataset of 3D shapes we train a set of the same number of MLPs, fitting each one on a single 3D shape. Every MLP is thus trained to approximate a continuous function that describes the represented shape, the nature of the function being chosen according to the discrete representation in which the shape is provided. We adopt MLPs with multiple hidden layers of the same dimension as done in (Sitzmann et al., 2020b;a; Dupont et al., 2021a; Strümpler et al., 2021; Zhang et al., 2021), interleaved by the sine activation function, as proposed in (Sitzmann et al., 2020b), to enhance the capability of the MLPs to fit the high frequency details of the input signal.

In its general formulation, an INR can be used to fit a continuous function $f : \mathbb{R}^{in} \to \mathbb{R}^{out}$. To do so, a training set composed of $N$ points $\mathbf{x}_i \in \mathbb{R}^{in}$ with $i = 1, 2, ..., N$, paired with values $\mathbf{y}_i = f(\mathbf{x}_i) \in \mathbb{R}^{out}$, is exploited to find the optimal parameters $\theta^*$ for the MLP that implements the INR, by solving the optimization problem:

$$\theta^* = \arg \min_{\theta} \frac{1}{N} \sum_{i=1}^{N} \ell(\mathbf{y}_i, f_\theta(\mathbf{x}_i)), \tag{1}$$

where $f_\theta$ represents the function $f$ approximated by the MLP with parameters $\theta$ and $\ell$ is a loss function that computes the error between predicted and ground-truth values.

The output value $f_\theta(\mathbf{x}_i)$ is computed as a series of linear transformations, each one followed by a non-linear activation function (*i.e.*, the sine function in our case), except the last one. Considering a MLP $m$, the mapping between its layers $L-1$ and $L$ consists in a linear transformation that maps the values $\mathbf{h}_m^{L-1} \in \mathbb{R}^{D_{L-1}}$ from the layer $L-1$ into the values $\mathbf{h}_m^L = \phi(\mathbf{W}_m^L \mathbf{h}_m^{L-1} + \mathbf{b}_m^L) \in \mathbb{R}^{D_L}$ of the layer $L$, with $\mathbf{W}_m^L$ being the matrix of weights $\in \mathbb{R}^{D_L \times D_{L-1}}$, $\mathbf{b}_m^L$ being the biases vector $\in \mathbb{R}^{D_L}$, and $\phi(\cdot)$ the non-linearity (Sitzmann et al., 2020b). If we now consider $M$ MLPs used to fit $M$ different INRs and the mapping between the layers $L-1$ and $L$, we can easily compute such mapping simultaneously for all the MLPs on modern GPUs thanks to tensor programming frameworks. The mapping consists indeed in a straightforward tensor contraction operation, where the values $\mathbf{h}_{L-1} \in \mathbb{R}^{M \times D_{L-1}}$ of the layer $L-1$ are mapped to the values $\mathbf{h}_L = \mathbf{W}_L \mathbf{h}_{L-1} + \mathbf{b}_L \in \mathbb{R}^{M \times D_L}$ of the layer $L$, with $\mathbf{W}^L \in \mathbb{R}^{M \times D_L \times D_{L-1}}$ and $\mathbf{b}^L \in \mathbb{R}^{M \times D_L}$. Extending this formulation to all the layers of the chosen MLP architecture allows to fit multiple INRs in parallel.

In the following, we describe how we train MLPs to obtain INRs starting from point clouds, triangle meshes and voxel grids.

**Point clouds.** The INR for a 3D shape represented by a point cloud $\mathcal{P}$ encodes the *unsigned distance function* (*udf*) of the point cloud $\mathcal{P}$. Given a point $\mathbf{p} \in \mathbb{R}^3$, the value $udf(\mathbf{p})$ is defined as $\min_{q \in \mathcal{P}} \|\mathbf{p} - \mathbf{q}\|_2$, *i.e.*, the euclidean distance from $\mathbf{p}$ to the closest point $\mathbf{q}$ of the point cloud. After preparing a training set of $N$ points $\mathbf{x}_i \in \mathbb{R}^3$ with $i = 1, 2, ..., N$, coupled with their *udf* values $y_i \in \mathbb{R}$, the INR of the underlying 3D shape is obtained by training a MLP to regress correctly the *udf* values, with the learning objective:

$$\mathcal{L}_{mse} = \frac{1}{N} \sum_{i=1}^{N} (y_i - f_\theta(\mathbf{x}_i))^2, \tag{2}$$

that consists in the mean squared error between ground-truth values $y_i$ and the predictions by the MLP $f_\theta(\mathbf{x}_i)$. An alternative objective is converting the *udf* values $y_i$ into values $y_i^{bce}$ continuously spanned in the range $[0, 1]$, with 0 and 1 representing respectively the predefined maximum distance

from the surface and the surface level set (*i.e.*, distance equal to zero). Then, the MLP optimizes the binary cross entropy between such labels and the predicted values, defined as:

$$\mathcal{L}_{bce} = -\frac{1}{N} \sum_{i=1}^{N} y_i^{bce} log(\hat{y}_i) + (1 - y_i^{bce}) log(1 - \hat{y}_i), \tag{3}$$

where $\hat{y}_i = \sigma(f_\theta(\mathbf{x}_i))$, with $\sigma$ representing the sigmoid function. In our experiments, we found empirically that this second learning objective leads to faster convergence and more accurate INRs, and we decided to adopt this formulation when producing INRs from point clouds.

**Triangle meshes.** Triangle meshes are usually adopted to represent closed surfaces. This provides an additional information compared to the point clouds case, since the 3D space can be divided into the portion contained *inside* and *outside* the closed surface. Thus, the INR of a closed 3D surface represented by a triangle mesh can be obtained by fitting the *signed distance function (sdf)* to the surface defined by the mesh. Given a point $\mathbf{p} \in \mathbb{R}^3$, the value $sdf(\mathbf{p})$ is defined as the euclidean distance from $\mathbf{p}$ to the closest point of the surface, with positive sign if $\mathbf{p}$ is outside the shape and negative sign otherwise. Similarly to the point clouds case, an INR for a 3D shape represented by a triangle mesh can be obtained by pursuing the learning objective presented in Eq. (2), using a training set composed of 3D points paired with their $sdf$ values. However, it possible to adopt a learning objective based on the binary cross entropy loss reported in Eq. (3) also for triangle meshes, and we empirically observed the same benefits. Hence, we adopt it also when fitting INRs on meshes. In this case, the $sdf$ values $y_i$ are converted into values $y_i^{bce} \in [0, 1]$, with 0 and 1 representing respectively the predefined maximum distance *inside* and *outside* the shape, *i.e.*, 0.5 represents the surface level set.

**Voxel grids.** A voxel grid is a 3D grid of $V^3$ cubes marked with label 1, if the cube is occupied, and label 0 otherwise. In order to fit an INR on voxels, it is possible to learn to regress the *occupancy function (occ)* of the grid itself. The training set, in this case, contains $V^3$ 3D points that corresponds to the centroids of the cubes that compose the voxel grid. Being each of such points $\mathbf{x}_i$ associated to a 0-1 label $y_i$, it is straightforward to use a binary classification objective to train the MLP that implement the desired INR. More specifically, we adopt the learning objective defined as:

$$\mathcal{L}_{focal} = -\frac{1}{N} \sum_{i=1}^{N} \alpha(1 - \hat{y}_i)^\gamma y_i log(\hat{y}_i) + (1 - \alpha)\hat{y}_i^\gamma (1 - y_i) log(1 - \hat{y}_i), \tag{4}$$

where $\hat{y}_i = \sigma(f_\theta(\mathbf{x}_i))$, while $\alpha$ and $\gamma$ are respectively the balancing parameter and the focusing parameter of the focal loss proposed in (Lin et al., 2017). We deploy a focal loss to alleviate the imbalance between the number of occupied and empty voxels.

## C    RECONSTRUCTING DISCRETE REPRESENTATIONS FROM INRS

In this section we discuss how it is possible to sample 3D discrete representations from INRs, which could be necessary to process the underlying shapes with algorithms that need an explicit surface (*e.g.*, Computational Fluid Dynamics (Baque et al., 2018; Toal & Keane, 2011; Umetani & Bickel, 2018)) or simply for visualization purposes.

**Point clouds from $udf$.** To sample a dense point cloud from an INR fitted on its $udf$, we use a slightly modified version of the algorithm proposed in (Chibane et al., 2020). The basic idea is to query the $udf$ with points scattered all over the considered portion of the 3D space, *projecting* such points onto the isosurface according to the predicted $udf$ values. In order to do that, let us define $f_\theta$ as the $udf$ approximated by the INR with parameters $\theta$. Given a point $\mathbf{p} \in \mathbb{R}^3$, it can be projected onto the isosurface by computing its updated position $\mathbf{p_s}$ as:

$$\mathbf{p_s} = \mathbf{p} - f_\theta(\mathbf{p}) \cdot \frac{\nabla_p f_\theta(\mathbf{p})}{\|\nabla_p f_\theta(\mathbf{p})\|}. \tag{5}$$

This can be intuitively understood by considering that the negative gradient of the $udf$ indicates the direction of maximum decrease of the distance from the surface, pointing towards the closest point on it. Eq. (5), thus, can be interpreted as moving $\mathbf{p}$ along the direction of maximum decrease of the $udf$ of a quantity defined by the value of the $udf$ itself in $\mathbf{p}$, reaching the point $\mathbf{p}_s$ on the

surface. One must consider, though, that $f_\theta$ is only an approximation of the real $udf$, which leads to two considerations. On a first note, the gradient of $f_\theta$ must be normalized (as done in Eq. (5)), while the gradient of the real $udf$ has norm equal to 1 everywhere except on the surface. Secondly, the predicted $udf$ value can be imprecise, implying that $\mathbf{p}$ can still be distant from the surface after moving it according Eq. (5). To address the second issue, the 3D position of $p_s$ is refined repeating the update described in Eq. (5) several times. Indeed, after each update, the point gets closer and closer to the surface, where the values approximated by $f_\theta$ are more accurate, implying that the last updates should successfully place the point on the isosurface. Given an INR fitted on the $udf$ of a point cloud, the overall algorithm to sample a dense point cloud from it is composed of the following steps. Firstly, we prepare a set of points scattered uniformly in the considered portion of the 3D space and we predict their $udf$ value with the given INR. Then we filter out points whose predicted $udf$ is greater than a fixed threshold (0.05 in our experiments). For the remaining points, we update their coordinates iteratively with Eq. (5) (we found 5 updates to be enough). Finally, we repeat the whole procedure until the reconstructed point cloud counts the desired number of points.

**Triangle meshes from** $sdf$**.** An INR fitted on the $sdf$ computed from a triangle mesh allows to reconstruct the mesh by means of the Marching Cubes algorithm (Lorensen & Cline, 1987). We refer the reader to the original paper for a detailed description of the method, but we report here a short presentation of the main steps, for the sake of completeness. Marching Cubes explores the considered 3D space by querying the $sdf$ with 8 locations at a time, that are the vertices of an arbitrarily small imaginary cube. The whole procedure involves *marching* from one cube to the other, until the whole desired portion of the 3D space has been covered. For each cube, the algorithm determines the triangles needed to model the portion of the isosurface that passes through it. Then, the triangles defined for all the cubes are fused together to obtain the reconstructed surface. In order to determine how many triangles are needed for a single cube and how to place them, for each pair of neighbouring vertices of the cube, their $sdf$ values are computed and one triangle vertex is placed between them if such values have opposite sign. Considering that the number of possible combinations of the $sdf$ signs at the cube vertices is limited, it is possible to build a look-up table to retrieve the triangles configuration for the cube starting from the $sdf$ signs at its eight vertices, combined in a 8-bit integer and used as key for the look-up table. After the triangles configuration for a cube has been retrieved, the vertices of the triangles are placed on the edges connecting the cube vertices, computing their exact position by linearly interpolating the two $sdf$ values that are connected by each edge.

**Voxel grids from** $occ$**.** In order to reconstruct voxel grids from INRs, we adopt a straightforward procedure. Each INR has been trained to predict the probability of a certain voxel to be occupied, when queried with the 3D coordinates of the voxel centroid. Thus, a first step to reconstruct the fitted voxels consists in creating a grid of the desired resolution $V$. Then, the INR is queried with the $V^3$ centroids of the grid and predicts an occupancy probability for each of them. Finally, we consider as occupied only voxels whose predicted probability is greater than a fixed threshold, which we set to 0.4, as we found empirically that it allows for a good trade-off between scattered and over-filled reconstructions.

## D `inr2vec` ENCODER AND DECODER ARCHITECTURES

In this section, we describe the architecture of `inr2vec` encoder, along with the one of the implicit decoder used to train it (see Section 3).

**Encoder.** `inr2vec` encoder, detailed in Fig. 12, consists in a series of linear transformations, that maps the input INR weights into features with higher dimensionality, before applying max pooling to obtain a compact embedding. More specifically, the input weights are rearranged in a matrix with shape $L(H + 1) \times H$, where $H$ is the number of nodes in the hidden layers of the MLP that implements the input INR and $L$ is the number of linear transformations between such hidden layers (*i.e.*, the MLP has $L + 1$ hidden layers). The matrix is obtained by stacking $L$ matrices (one for each linear transformation), each one with shape $(H + 1) \times H$, being composed of a matrix of weights with shape $H \times H$ and a row of $H$ biases. In our setting, each MLP has 4 hidden layers with 512 nodes: the final matrix in input to `inr2vec` encoder has shape $3 \cdot (512 + 1) \times 512 = 1539 \times 512$. In the current implementation, the four linear mappings of the encoder transform each row of the input matrix into features with size 512, 512, 1024 and 1024, obtaining, at each step, features matrices with shape $1539 \times 512$, $1539 \times 512$, $1539 \times 1024$ and $1539 \times 1024$. Finally, the encoder applies

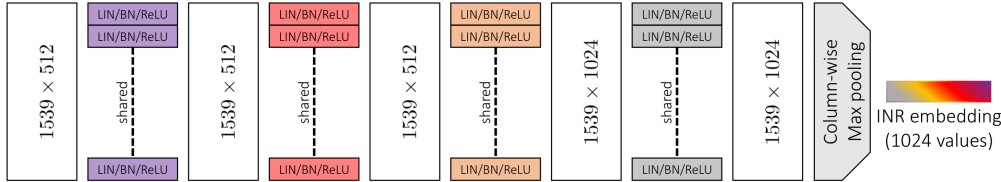

Figure 12: `inr2vec` **encoder.** With a series of linear transformations and a final column-wise max pooling, the encoder maps the input weights matrix into a compact embedding. LIN/BN/ReLU stands for a linear transformation, followed by batch normalization and ReLU activation function.

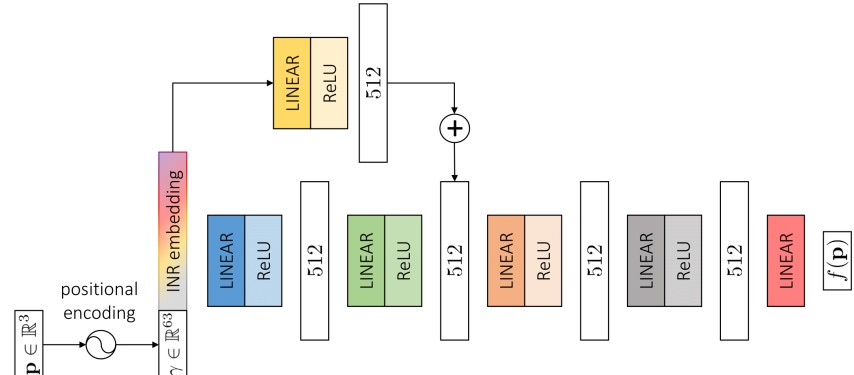

Figure 13: `inr2vec` **decoder.** Our framework is trained with an implicit decoder, that maps an INR embedding concatenated with a 3D query into the value of the implicit function at the query coordinates.

column-wise max pooling to compress the final matrix into a single compact embedding composed of 1024 values. Between the linear mappings of the encoder, we adopt 1D batch normalization and ReLU activation functions.

**Decoder.** The implicit decoder that we adopt to train `inr2vec` is presented in Fig. 13. We designed it taking inspiration from (Park et al., 2019), since we need a decoder capable of reproducing the implicit function of input INR when conditioned on the embedding obtained by the encoder. Thus, the decoder takes in input the concatenation of the INR embedding with the coordinates of a given 3D query. We adopt the positional encoding proposed in (Mildenhall et al., 2020) to embed the input 3D coordinates into a higher dimensional space to enhance the capability of the decoder to capture the high frequency variations of the input data. The query 3D coordinates are mapped into 63 values that, concatenated with the 1024 values that compose the INR embedding, result in a vector with 1087 values as input for `inr2vec` decoder. Internally, the implicit decoder is composed of 4 hidden layers with 512 nodes and of a skip connection that projects the input 1087 values into a vector of 512 elements, that are summed to the features of the second hidden layer before being fed to the transformation that bridges the second and the third hidden layers. Finally, the features of the last hidden layer are mapped to a single output, which is compared to the ground-truth associated with the input 3D query to compute the loss. Each linear transformation of the decoder, except the output one, is paired with the ReLU activation function.

## E    MOTIVATION BEHIND `inr2vec` ENCODER DESIGN

We designed `inr2vec` encoder with the goal of obtaining a good scalability in terms of memory occupation. Indeed, a naive solution to process the weights of an input INR would consist in an MLP encoder mapping the flattened vector of weights to the embedding of the desired dimension. However, such approach would require a huge amount of memory resources, since an input INR of 4 layers of 512 neurons would have approximately 800K parameters. Thus, an MLP encoder going from 800K parameters to an embedding space of size 1024 would already have a totality of ~800M parameters. We report in Table 5 a detailed analysis of the parameters of our encoder *w.r.t.* the ones of an MLP encoder by varying the input INR dimension. As we can notice the MLP encoder does

| INR hidden dim. | INR #layers | INR #params | #params inr2vec encoder | #params MLP encoder |
|---|---|---|---|---|
| 512 | 4 | ∼800K | ∼3M | ∼800M |
| 512 | 8 | ∼2M | ∼3M | ∼2B |
| 512 | 12 | ∼3M | ∼3M | ∼3B |
| 512 | 16 | ∼4M | ∼3M | ∼4B |
| 1024 | 4 | ∼3M | ∼3.5M | ∼3B |
| 1024 | 8 | ∼7M | ∼3.5M | ∼7.5B |
| 1024 | 12 | ∼11M | ∼3.5M | ∼12B |
| 1024 | 16 | ∼15M | ∼3.5M | ∼16B |

Table 5: **Number of parameters of** `inr2vec` **encoder.** Comparison between the number of parameters of `inr2vec` encoder and the number of parameters of a generic MLP encoder.

not scale well, making this kind of approach very expensive in practice, while `inr2vec` encoder scales gracefully to bigger input INRs.

## F  EXPERIMENTAL SETTINGS

We report here a detailed description of the settings adopted in our experiments.

**INRs fitting.** In every experiment reported in the paper, we fit INRs on 3D discrete representations using MLPs having 4 hidden layers with 512 nodes each. We implement MLPs using sine as a periodic activation function, as proposed in (Sitzmann et al., 2020b). The procedure adopted to fit a single MLP consists in querying it with 3D points sampled properly in the space surrounding the underlying shape. The MLP predicts a value for each query and it's trained by computing a loss function between the predicted value and the ground-truth value of the fitted implicit function (*i.e.*, $udf$ for point clouds, $sdf$ for meshes and $occ$ for voxels). The set of training queries is prepared according to different strategies, depending on the nature of the discrete representation being fitted. For voxel grids, the set of possible queries consists of the 3D coordinates of all the centroids of the grid. For point clouds and meshes, instead, queries are sampled with different densities in the volume containing the fitted shape: indeed, for each shape, we prepare 500K queries by taking 250K points close to the surface, 200K points at a medium-far distance for the surface, 25K far from the surface and other 25K scattered uniformly in the volume. The queries coordinates are computed by adding gaussian noise to the points of the fitted point cloud or to points sampled uniformly from the fitted mesh surface. More precisely, close queries are computed with noise sampled from the normal distribution $\mathcal{N}(0, 0.001)$, medium-far queries with noise from $\mathcal{N}(0, 0.01)$, far queries with noise from $\mathcal{N}(0, 0.1)$. The uniformly scattered queries are just computed by sampling each of their coordinates from the uniform distribution $\mathcal{U}(-1, 1)$, being the considered shapes normalized in such volume. As for the ground-truth values, for voxels they consist simply in the occupied/empty label of the voxel associated to the query. For point clouds, for each query we compute its $udf$ value by building a KDTree on the fitted point cloud and looking for the closest point to the considered query (we used the PyTorch3D (Ravi et al., 2020) implementation of the KDTree algorithm). For meshes, finally, we compute the $sdf$ of queries with the functions provided in the Open3D library (Zhou et al., 2018)[1]. For each of the considered modalities, at each step of the fitting procedure, we randomly sample 10K pairs of queries/ground-truth values from the precomputed ones, performing a total of 500 steps for each shape. Thanks to the procedure detailed in Appendix B, we are able to fit up to 16 multiple MLPs in parallel, using Adam optimizer (Kingma & Ba, 2014) with learning rate set to 1e-4. On a final note, we fixed the weights initialization of the MLPs to be always the same, as we observed empirically this to be key to convergence of `inr2vec`. This choice poses no limitation to the practical use of our framework and has also been adopted in recent works (Gropp et al., 2020; Sitzmann et al., 2020a).

`inr2vec` **training.** According to what is described in Section 3, during training our framework takes in input the weights of a given INR and is asked to reproduce the implicit function fitted by the INR on a set of predefined 3D queries. Such queries are prepared with the same strategies described in the previous paragraph and, similarly to what is done while fitting INRs, at each step the training loss for `inr2vec` is computed on 10K queries randomly sampled from a set of precomputed ones. In every experiment, we train `inr2vec` with AdamW optimizer (Loshchilov & Hutter, 2017), learning rate 1e-4 and weight decay 1e-2 for 300 epochs, one epoch corresponding to processing all the INRs

---

[1] http://www.open3d.org/docs/latest/tutorial/geometry/distance_queries.html

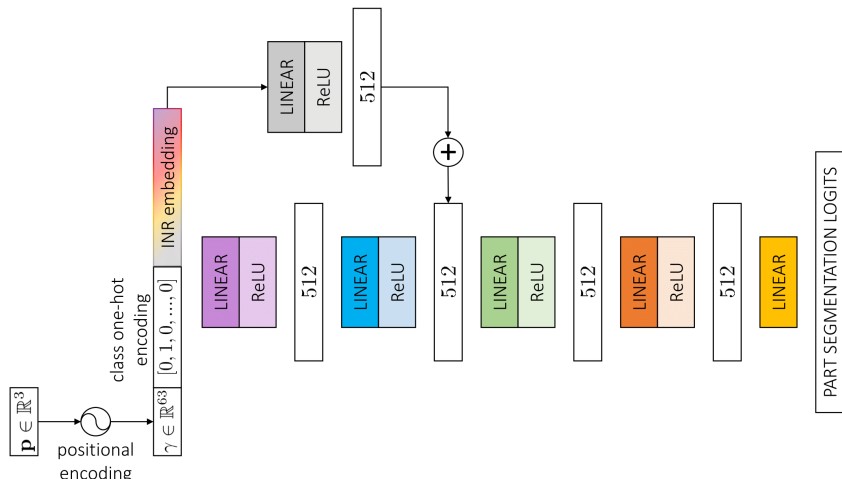

Figure 14: **Part segmentation decoder.** We train a decoder to predict the part segmentation label of a given 3D query when conditioned on the embedding of the input INR and on the one-hot encoding of the INR class.

that compose the considered dataset, processing at each training step a mini-batch of 16 INRs. During training, we select the best model by evaluating its reconstruction capability on a validation set of INRs. When training on INRs obtained from point clouds, we compare the ground-truth set of points with the ones reconstructed by `inr2vec` decoder. For voxels, we compare the input and the output grid by comparing the point clouds composed by the centroids corresponding to occupied voxels. As for what concerns meshes, we compare the clouds containing input and output vertices. In all cases, the reconstruction quality is evaluated by computing the Chamfer Distance between ground-truth and output point clouds, as defined in (Fan et al., 2017). See Appendix C of this document for details on how to sample discrete 3D representations from the implicit functions fitted by INRs and that `inr2vec` is trained to reproduce.

**Shape classification.** The classifier that we deploy on `inr2vec` embeddings is composed of three linear transformations, mapping sequentially the input embedding with 1024 features to vectors of size 512, 256 and, finally, to a vector with a number of values corresponding to the number of classes of the considered dataset. The final vector is then transformed to a probability distribution with the softmax function. We use 1D batch normalization and the ReLU activation function between the classifier linear transformations. In all experiments, the classifier is trained for 150 epochs, with AdamW optimizer (Loshchilov & Hutter, 2017) and weight decay 1e-2. The learning rate is scheduled according to the OneCycle strategy (Smith & Topin, 2019), with maximum learning rate set to 1e-4. At each training step, the classifier processes a mini-batch counting 256 embeddings. During training, we select the best model by computing the classification accuracy on a validation set of embeddings. The best model is used after training to compute the classification accuracy on the test set, obtaining the numbers reported in the tables of the paper.

**Point cloud part segmentation.** In order to tackle point cloud part segmentation starting from `inr2vec` embeddings, we adopt a decoder similar to the one that we use for reconstruction during `inr2vec` training. The part segmentation decoder, depicted in Fig. 14, is fed with the positional encoding of a 3D query together with the embedding of an input INR and predicts a $K$-dimensional vector of segmentation logits for the given query, with $K$ representing the total number of parts of all the $C$ available categories. Moreover, as done in previous work (Qi et al., 2017a;b; Wang et al., 2019b), we concatenate an additional $C$-dimensional vector to the input of the part segmentation decoder, conditioning the output of our decoder with the one-hot encoding of the input INR class. We conduct our experiments on the ShapeNet Part Segmentation dataset (Yi et al., 2016), that presents 16 categories labeled with two to five parts, for a total of 50 parts (*i.e.*, $C$=16 and $K$=50). According to a standard protocol (Qi et al., 2017a;b; Wang et al., 2019b), during training we compute the cross-entropy loss function on all the $K$ logits predicted by our decoder, while, at test time, the final prediction is obtained considering only the subset of parts belonging to the specific class of the input INR. The part segmentation decoder is trained with the original point clouds available in the ShapeNet

Part Segmentation dataset, where part labels are provided for each point of each cloud. At test time, though, we test both our decoder and the considered competitors on the point clouds reconstructed from the input INRs, since we want to simulate the scenario of 3D shapes being available exclusively in the form of INRs. Thus, the protocol to obtain a segmented point cloud starting from an input INR consists in reconstructing the cloud first (see Appendix C) and then in assigning a part label to each point of the reconstructed shape with our part segmentation decoder. When ground-truth labels are required to compute quantitative results, we obtain them by comparing the reconstructed cloud with the original one and assigning to each point of the reconstructed shape the part label of the closest point in the original cloud. Our part segmentation decoder is trained for 250 epochs with AdamW optimizer, OneCycle learning rate scheduling with maximum value set to 1e-4, weight decay equal to 1e-2 and mini-batches composed of 256 embeddings, each one paired with 3D queries from the original point clouds during training and from the ones reconstructed from the input INRs at test time. During training, we compute the class mIoU on the validation split and save the best model in order to compute the final metrics on the test set.

**Shape generation.** We perform unconditional shape generation by training Latent-GAN (Achlioptas et al., 2018) to generate embeddings indistinguishable from the ones produced by `inr2vec` on a given dataset. This approach allows us to train a shape generation framework with the very same architecture to generate embeddings representing INRs with different underlying implicit functions, such as $udf$ for the point clouds of ShapeNet10 and $sdf$ for the models of cars provided by (Mescheder et al., 2019). We conducted our experiments using the official implementation[2], setting all the hyperparameters to default. The generator network is implemented as a fully connected network with two layers and ReLU non linearity, that map an input noise vector with 128 values sampled from the normal distribution $\mathcal{N}(0, 0.2)$ to an intermediate hidden vector of the same dimension and then to the predicted embedding with 1024 values (we removed the final ReLU present in the original implementation). The discriminator is also a fully connected network, with three layers and ReLU non linearity. The first layer maps the embedding produced by the generator to a hidden vector with 256 values, which are then transformed by the second layer into a hidden vector with 512 values, that are finally used by the third layer, together with the sigmoid function, to predict the final score. According to the original implementation, we trained one separate Latent-GAN for each class of the considered datasets, using the Wasserstein objective with gradient penalty proposed in (Gulrajani et al., 2017) and training each model for 2000 epochs.

**Learning a mapping between `inr2vec` embedding spaces.** The transfer function between `inr2vec` embedding spaces is implemented as a simple fully connected network, with 8 linear layers interleaved by 1D batch norm and ReLU activation functions. All the hidden features produced by the linear transformations present the same dimension of the input embedding, *i.e.*, 1024 values. The final linear layer predicts the output embedding, which is compared with the target one with a standard L2 loss. We train the transfer network with AdamW optimizer, constant learning rate and weight decay both set to 1e-4, stopping the training upon convergence, which we measure by comparing the shapes reconstructed by the predicted embeddings with the ground-truth ones on a predetermined validation split. Such validation metrics are used also to save the best model during training, which is finally evaluated on the test set.

## G   IMPLEMENTATION, HARDWARE AND TIMINGS

We implemented our framework with the PyTorch library, performing all the experiments on a single NVIDIA 3090 RTX GPU. We created an augmented version of each considered dataset, in order to obtain roughly ∼100K INRs, whose fitting requires around 4 days in the current implementation. Training `inr2vec` requires another 48 hours, while all the networks adopted to perform the downstream tasks on `inr2vec` embeddings can be trained in few hours.

## H   TESTING ON ORIGINAL DISCRETE 3D REPRESENTATIONS

In the experiments "Shape classification" and "Point cloud part segmentation" presented in the paper, we evaluated the competitors on the 3D discrete representations reconstructed from the INRs fitted on the test sets of the considered datasets, since these would be the only data available at test

---

[2]`https://github.com/optas/latent_3d_points`

| Method | Point Cloud | | | Mesh | Voxels |
| --- | --- | --- | --- | --- | --- |
| | ModelNet40 | ShapeNet10 | ScanNet10 | Manifold40 | ShapeNet10 |
| PointNet (Qi et al., 2017a) | 88.8 | 94.7 | 72.8 | – | – |
| PointNet++ (Qi et al., 2017b) | 91.0 | 95.2 | 76.3 | – | – |
| DGCNN (Wang et al., 2019b) | 91.6 | 94.0 | 75.1 | – | – |
| MeshWalker (Lahav & Tal, 2020) | – | – | – | 90.6 | – |
| Conv3DNet (Maturana & Scherer, 2015) | – | – | – | – | 92.5 |
| `inr2vec` | 87.0 | 93.3 | 72.1 | 86.3 | 93.0 |

Table 6: **Shape classification results.** We report here shape classification results when testing on the original discrete representations of the test sets instead of reconstructing them from the input INRs.

| Method | instance mIoU | class mIoU | airplane | bag | cap | car | chair | earphone | guitar | knife | lamp | laptop | motor | mug | pistol | rocket | skateboard | table |
| --- | --- | --- | --- | --- | --- | --- | --- | --- | --- | --- | --- | --- | --- | --- | --- | --- | --- | --- |
| PointNet (Qi et al., 2017a) | 83.0 | 78.8 | 80.5 | 77.9 | 78.3 | 74.4 | 89.0 | 68.3 | 90.1 | 82.2 | 80.7 | 94.7 | 63.1 | 91.7 | 79.3 | 58.2 | 72.7 | 81.0 |
| PointNet++ (Qi et al., 2017b) | 84.4 | 82.8 | 81.7 | 86.5 | 85.2 | 78.6 | 90.2 | 77.9 | 91.2 | 84.4 | 83.2 | 95.4 | 72.0 | 94.6 | 83.3 | 64.2 | 75.6 | 80.9 |
| DGCNN (Wang et al., 2019b) | 84.3 | 81.4 | 81.6 | 82.2 | 80.9 | 75.7 | 90.7 | 80.9 | 90.2 | 86.9 | 82.6 | 94.8 | 64.8 | 92.8 | 81.0 | 60.6 | 74.7 | 81.8 |
| `inr2vec` | 80.5 | 71.1 | 79.5 | 72.9 | 72.3 | 70.7 | 87.4 | 64.1 | 89.4 | 81.6 | 76.5 | 94.5 | 59.3 | 92.4 | 78.4 | 53.5 | 67.5 | 77.3 |

Table 7: **Part segmentation results.** In this table we present part segmentation results when testing on the original discrete representations of the test sets instead of reconstructing them from the input INRs. We report the IoU for each class, the mean IoU over all the classes (class mIoU) and the mean IoU over all the instances (instance mIoU).

time in a scenario where INRs are used to store and communicate 3D data. For completeness, we report here the scores achieved by the baselines when tested on the original discrete representations, without reconstructing them from the input INRs. Such results are presented in Table 6 for shape classification and in Table 7 for part segmentation. We report in the tables also the results obtained with our framework: they are the same reported in Table 2 for what concerns shape classification, since our classifier processes exclusively `inr2vec` embeddings, while they are different for part segmentation, as we use as query points for our segmentation decoder those from the discrete point clouds reconstructed from input INRs in Table 3 while we use those from the original point clouds in the experiment reported here, as done for the competitors. The results reported in the tables show limited differences, either positive or negative, with the ones presented in Section 4, mostly within the range of variations due to the inherent stochasticity of training. There are few larger differences, like DGCNN on ModelNet40 (+1.7 when tested on the original discrete representations) or on ScanNet (-1.1 when tested on the original discrete representations), whose difference in sign however suggests neither of the two settings is clearly superior to the other.

# I   ALTERNATIVE ARCHITECTURE FOR `inr2vec`

As reported in Section 3, `inr2vec` encoder takes in input the weights of an INR reshaped in a suitable way, discarding the parameters of the first and of the last layers. In this section we consider the possibility of processing all the weights of the input INR, including the input/output ones. To this end, one must properly arrange the input/output parameters since they feature different dimensionality from the ones of the hidden layers and cannot be seamlessly stacked together with them. More specifically, the first layer of an INR consists in a matrix of weights $\mathbf{W}_{in} \in \mathbb{R}^{H \times D}$ and in a vector of biases $\mathbf{b}_{in} \in \mathbb{R}^{H \times 1}$, with $H$ being the dimension of the hidden features of the INR and $D$ being the dimension of the inputs (*i.e.*, 3 in our case of 3D coordinates). The output layer, instead, is responsible of transforming the final vector of hidden features to the predicted output, which is always a single value in the cases considered in our experiments (*i.e.*, $udf$, $sdf$ and $occ$). Thus, the last layer presents a matrix of weights $\mathbf{W}_{out} \in \mathbb{R}^{1 \times H}$ and single bias $\mathbf{b}_{out}$. In order to include the input/output parameters in the matrix $\mathbf{P}$ presented in input to `inr2vec` encoder (see Section 3),

| Architecture | F-Score ↑ | CD (mm) ↓ |
| --- | --- | --- |
| hidden layers | 57.41 | 3.1 |
| all layers | 56.76 | 3.1 |

Table 8: **Quantitative comparison between alternative `inr2vec` architectures**. We compare the reconstruction capability of `inr2vec` when processing only the weights of the hidden layers ("hidden layers") or all the weights ("all layers") of the input INRs.

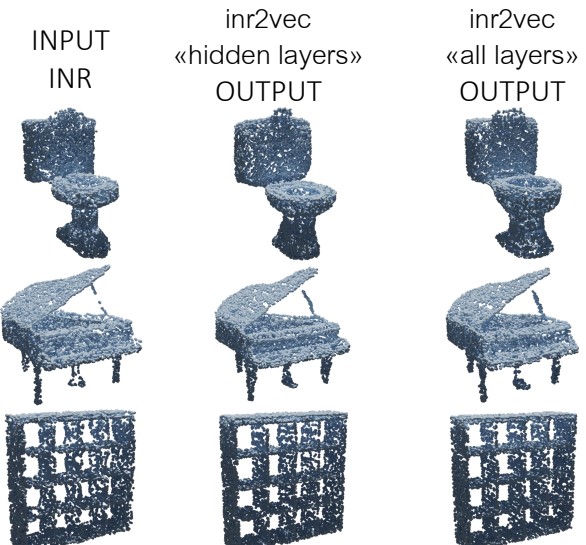

Figure 15: **Qualitative comparison between alternative** `inr2vec` **architectures**. We compare the reconstruction capability of `inr2vec` when processing only the weights of the hidden layers ("hidden layers") or all the weights ("all layers") of the input INRs.

$\mathbf{W}_{in}$ needs to be transposed, obtaining a matrix with shape $3 \times H$, $\mathbf{b}_{in}$ is transposed as done also for the biases of all the other layers, $\mathbf{W}_{out}$ doesn't need any manipulation and we decided to repeat the single-valued $\mathbf{b}_{out}$ $H$ times. In this section, we compare the formulation presented in Section 3 (reported as "hidden layers") with the one proposed here (reported as "all layers"), looking at the reconstruction capabilities of the two variants of our framework when trained on ModelNet40. In Table 8, we report both the F-score (Tatarchenko et al., 2019) and the Chamfer Distance (CD) (Fan et al., 2017) between the clouds used to obtain the INRs presented in input to `inr2vec` and the ones reconstructed from `inr2vec` embeddings, while in Fig. 15 we show the same comparison from a qualitative perspective. Results show that processing all the INR weights doesn't produce any significant difference *w.r.t.* ingesting only the weights of the hidden layers. However, the latter variant provides a slight advantage in terms of F-score, simplicity and processing time, motivating our choice to adopt it as formulation for `inr2vec`.

## J  ADDITIONAL QUALITITATIVE RESULTS

We report here additional qualitative results. In Fig. 16, Fig. 17 and Fig. 18 we show some comparisons between the discrete shapes reconstructed from input INRs and the ones reconstructed from `inr2vec` embeddings. Fig. 19, Fig. 20 and Fig. 21 present smooth interpolations between `inr2vec` embeddings. In Fig. 22 and Fig. 23 we propose additional qualitative results for the point cloud retrieval experiments. Fig. 24 shows qualitative results for point cloud part segmentation for all the classes of the ShapeNet Part Segmentation dataset. Fig. 25, Fig. 26 and Fig. 27 report shapes generated with Latent-GANs (Achlioptas et al., 2018) trained on `inr2vec` embeddings. Finally, Fig. 28 and Fig. 29 present additional qualitative results for the experiments of point cloud completion and surface reconstruction, tackled by learning a mapping between `inr2vec` latent spaces.

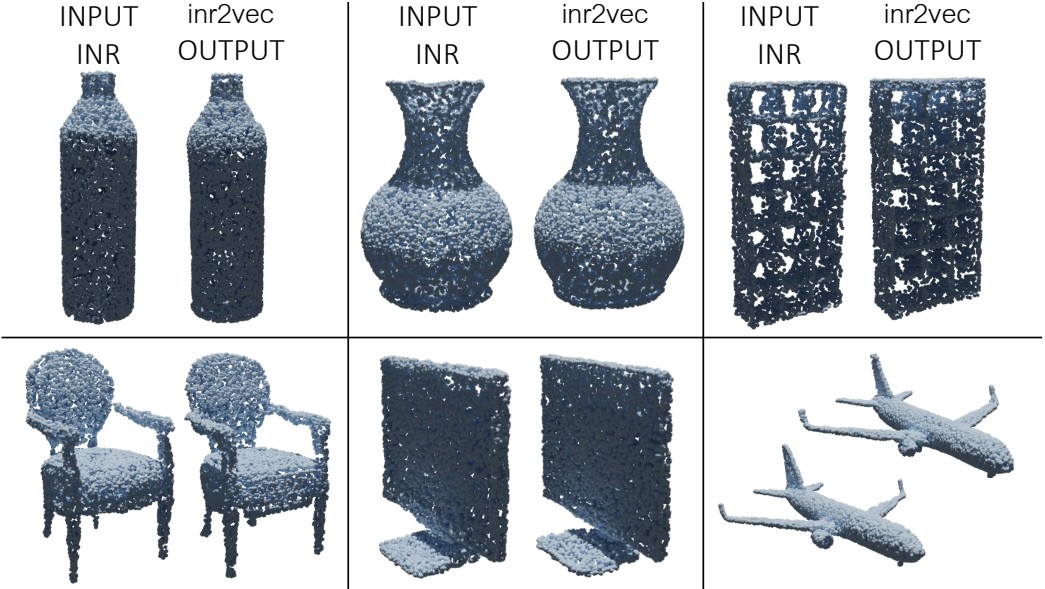

Figure 16: `inr2vec` **reconstructions when dealing with INRs fitted on point clouds.** Comparison between discrete shapes reconstructed from the INRs presented in input to `inr2vec` ("INPUT INR") and the ones reconstructed from `inr2vec` embeddings ("`inr2vec` OUTPUT").

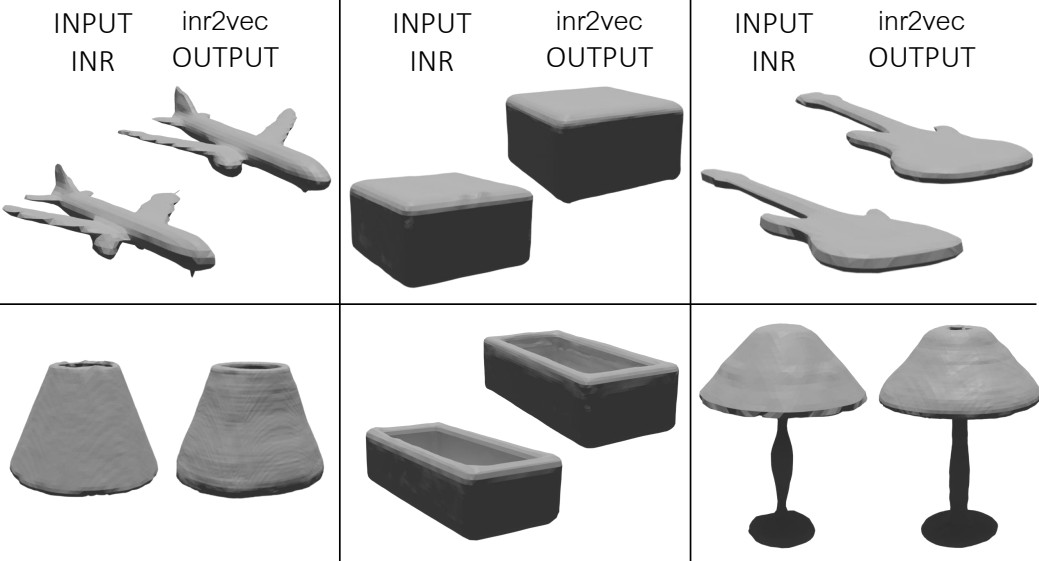

Figure 17: `inr2vec` **reconstructions when dealing with INRs fitted on meshes.** Comparison between discrete shapes reconstructed from the INRs presented in input to `inr2vec` ("INPUT INR") and the ones reconstructed from `inr2vec` embeddings ("`inr2vec` OUTPUT").

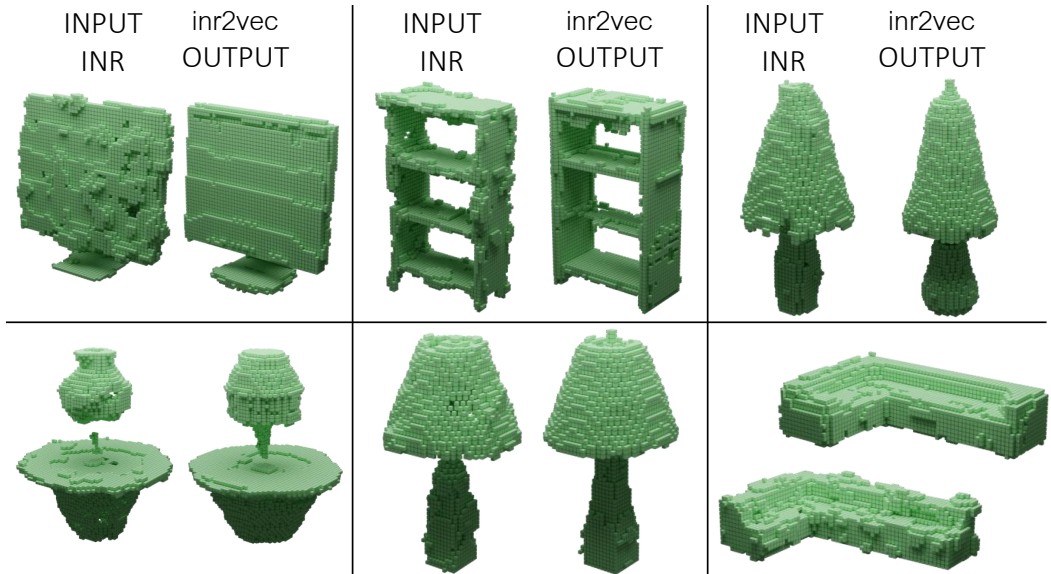

Figure 18: `inr2vec` **reconstructions when dealing with INRs fitted on voxel grids.** Comparison between discrete shapes reconstructed from the INRs presented in input to `inr2vec` ("INPUT INR") and the ones reconstructed from `inr2vec` embeddings ("`inr2vec` OUTPUT").

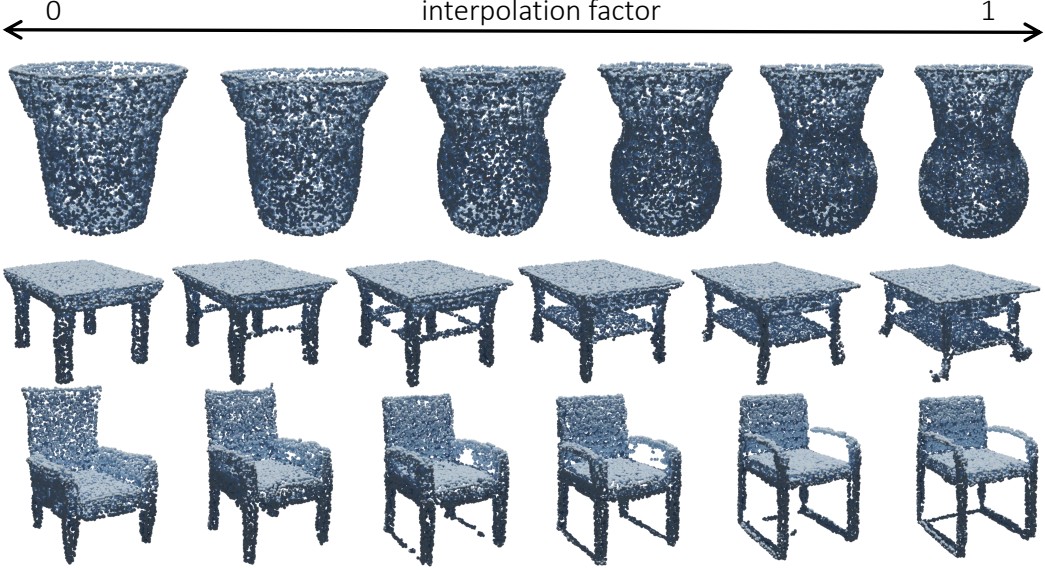

Figure 19: `inr2vec` **latent space interpolation.** Given two `inr2vec` embeddings obtained from INRs fitted on point clouds, it is possible to linearly interpolate between them, producing new embeddings that represent unseen INRs of plausible shapes.

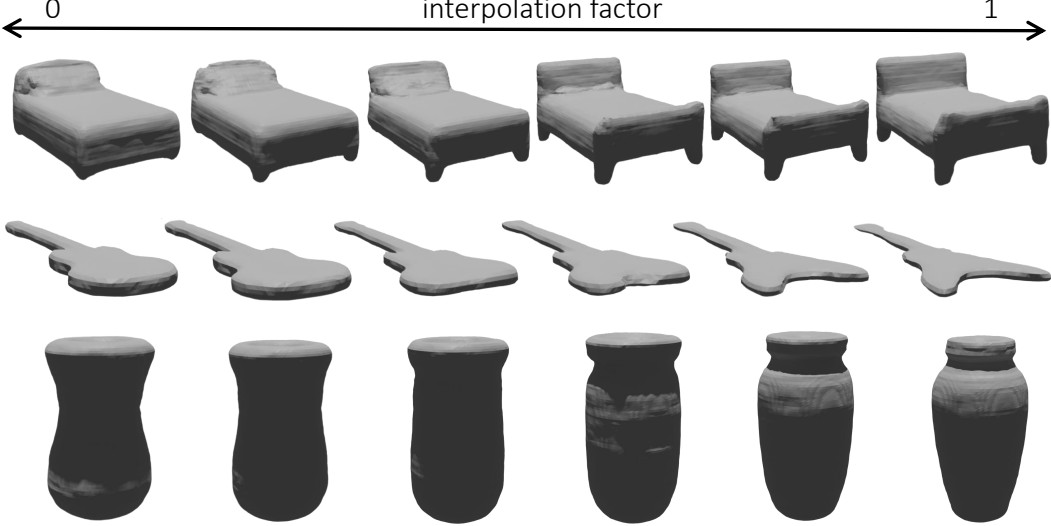

Figure 20: `inr2vec` **latent space interpolation.** Given two `inr2vec` embeddings obtained from INRs fitted on meshes, it is possible to linearly interpolate between them, producing new embeddings that represent unseen INRs of plausible shapes.

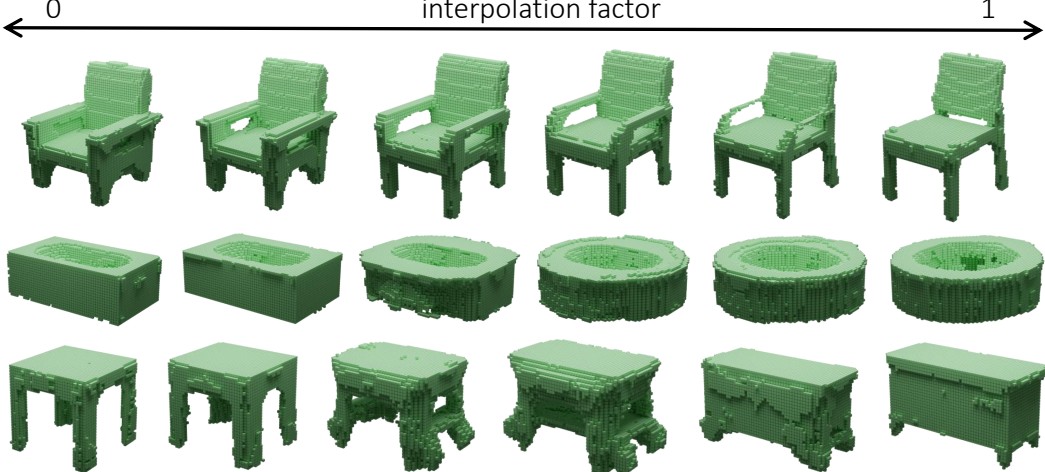

Figure 21: `inr2vec` **latent space interpolation.** Given two `inr2vec` embeddings obtained from INRs fitted on voxel grids, it is possible to linearly interpolate between them, producing new embeddings that represent unseen INRs of plausible shapes.

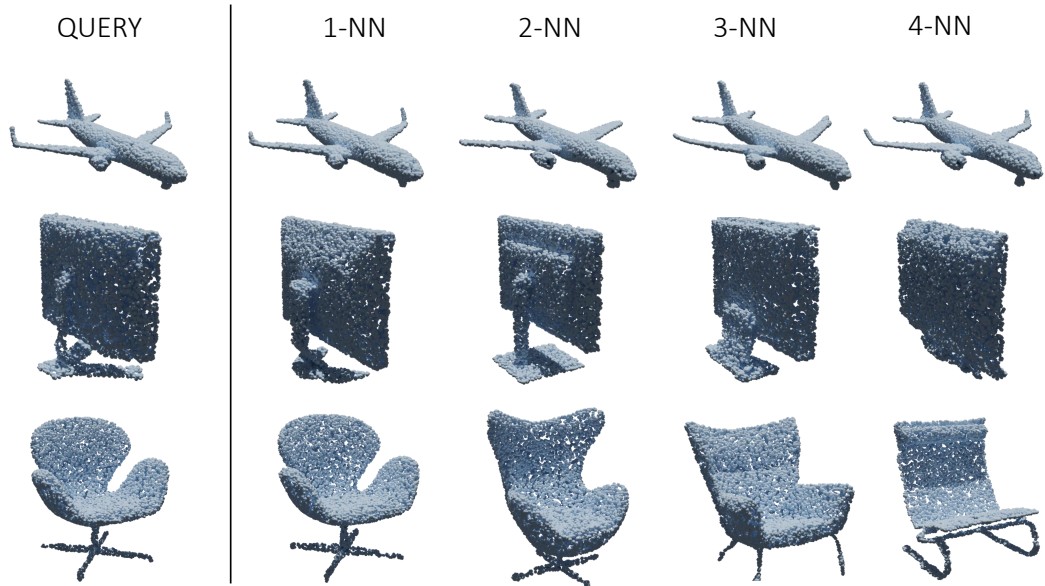

Figure 22: **Point cloud retrieval (ModelNet40).** Qualitative results of the point cloud retrieval experiment conducted on inr2vec latent space. We show the discrete shape reconstructed from the query INR on the left and the discrete clouds reconstructed from the closest inr2vec embeddings in the columns 2-5.

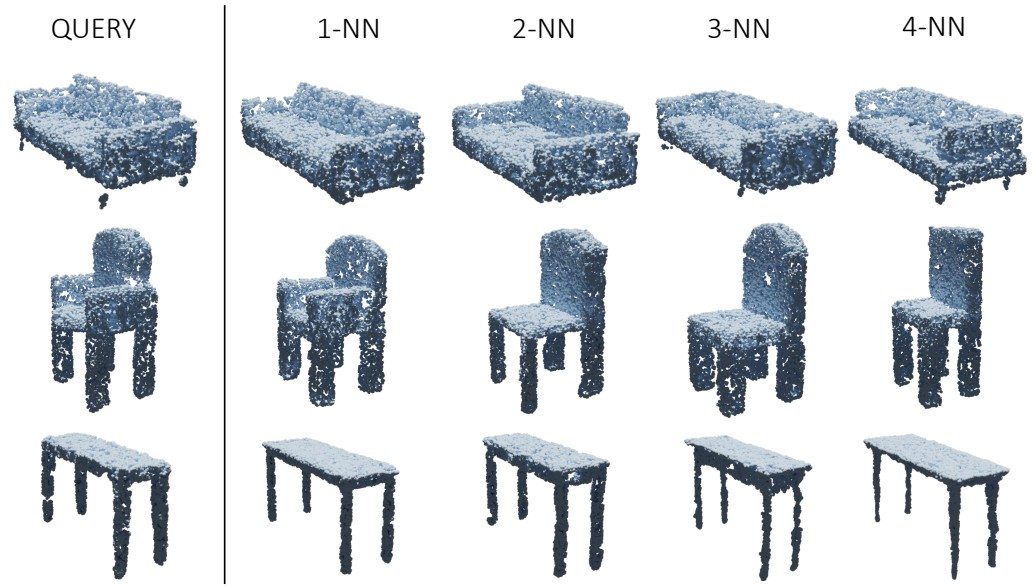

Figure 23: **Point cloud retrieval (ShapeNet10).** Qualitative results of the point cloud retrieval experiment conducted on inr2vec latent space. We show the discrete shape reconstructed from the query INR on the left and the discrete clouds reconstructed from the closest inr2vec embeddings in the columns 2-5.

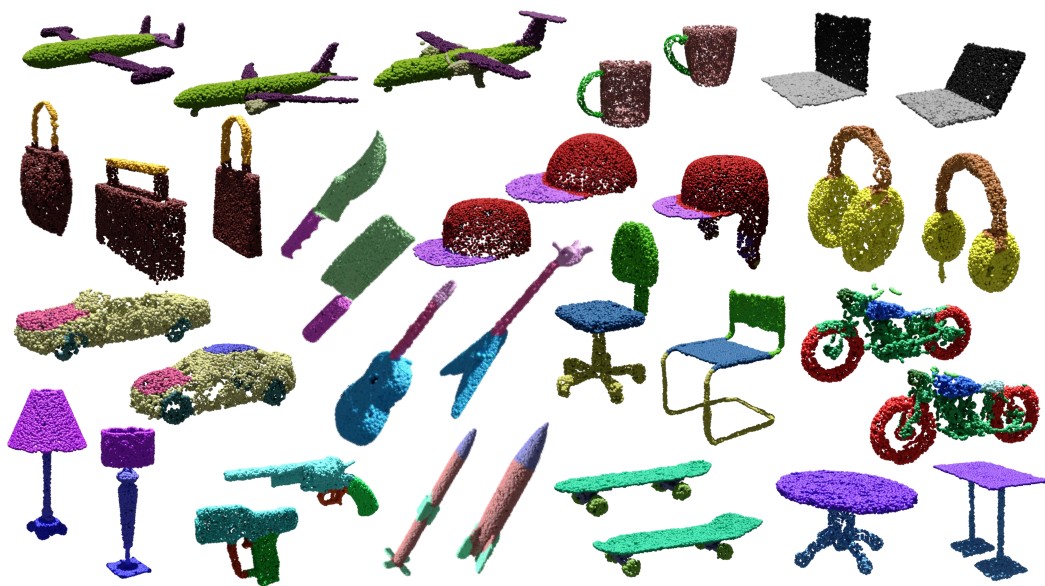

Figure 24: **Point cloud part segmentation.** Qualitative results of the part segmentation experiment conducted with our segmentation decoder conditioned on `inr2vec` embeddings.

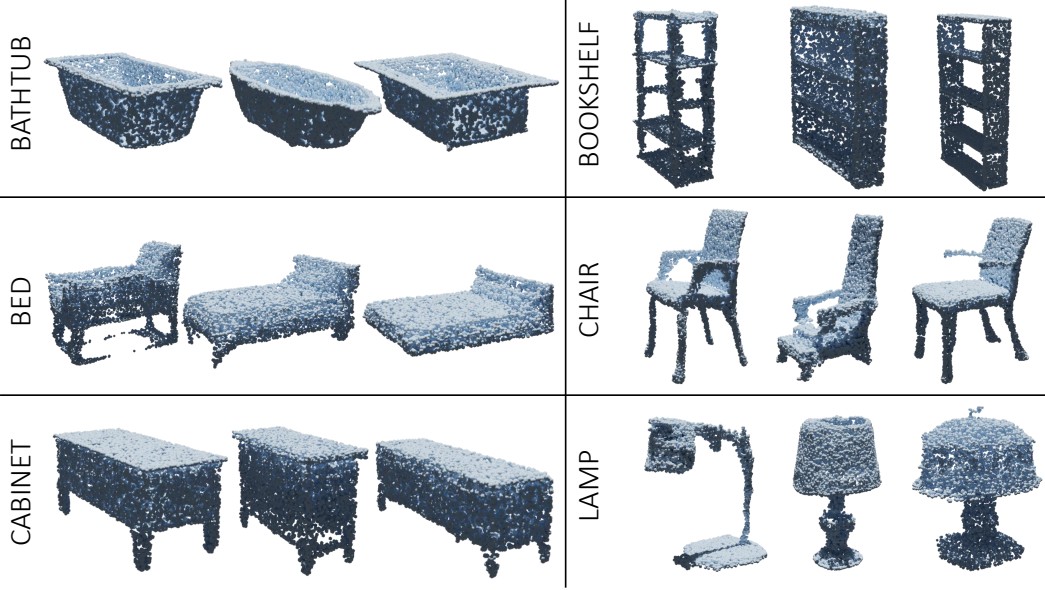

Figure 25: **Shape generation (point clouds).** We show point clouds reconstructed from embeddings generated by a Latent-GAN trained on `inr2vec` embeddings (one model for each class).

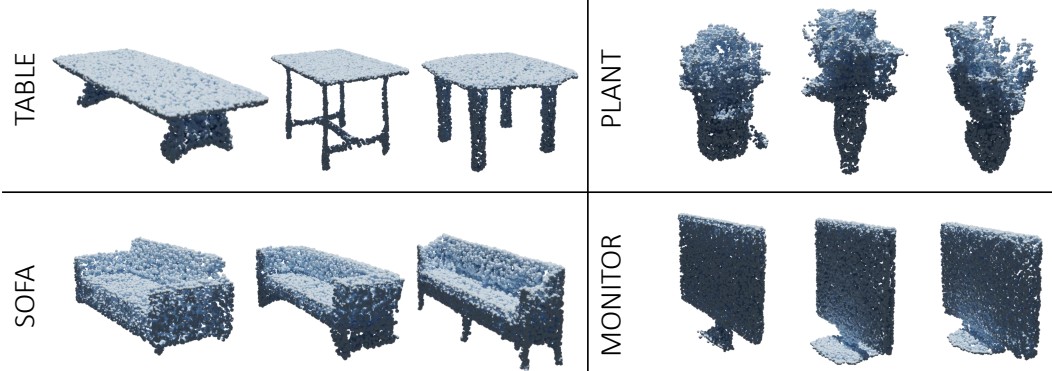

Figure 26: **Shape generation (point clouds).** We show point clouds reconstructed from embeddings generated by a Latent-GAN trained on `inr2vec` embeddings (one model for each class).

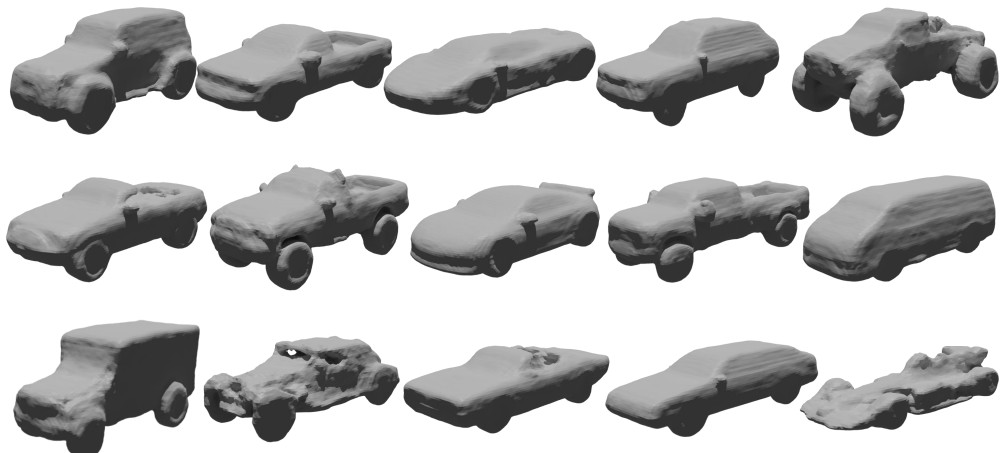

Figure 27: **Shape generation (meshes).** We show meshes reconstructed from embeddings generated by a Latent-GAN trained on `inr2vec` embeddings.

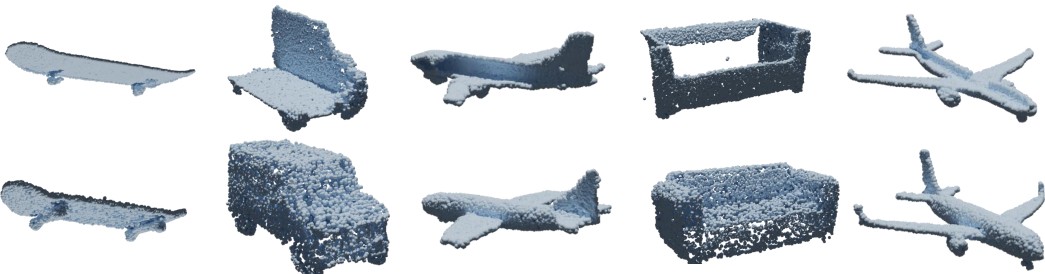

Figure 28: **Point cloud completion.** Qualitative results of the point cloud completion experiment conducted with a transfer network that learns a mapping between `inr2vec` latent spaces.

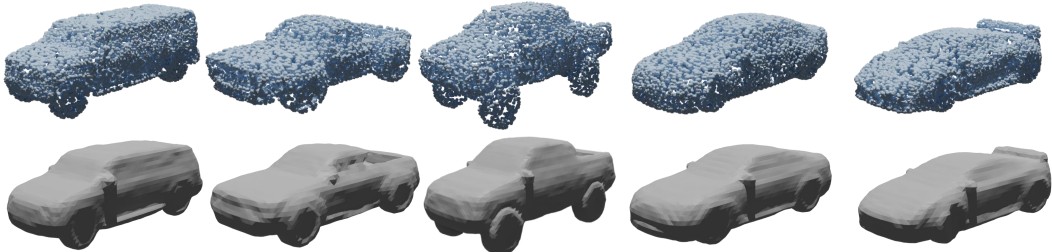

Figure 29: **Surface reconstruction.** Qualitative results of the surface reconstruction experiment conducted with a transfer network that learns a mapping between `inr2vec` latent spaces.

## K EFFECTIVENESS OF USING THE SAME INITIALIZATION FOR INRS

The need to align the multitude of INRs that can approximate a given shape is a challenging research problem that has to be dealt with when using INRs as input data. We empirically found that fixing the weights initialization to a shared random vector across INRs is a viable and simple solution to this problem.

We report here an experiment to assess if order of data, or other sources of randomness arising while fitting INRs, do affect the repeatability of the embeddings computed by `inr2vec`. We fitted 10 INRs on the same discrete shape for 4 different chairs, *i.e.*, 40 INRs in total. Then, we embed all of them with the pretrained `inr2vec` encoder and compute the L2 distance between all pairs of embeddings. The block structure of the resulting distance matrix (see Fig. 30) highlights how, under the assumption of shared initialization and hyperparameters, `inr2vec` is repeatable across multiple fittings.

Seeking for a proof with a stronger theoretical foundation, we turn our attention to the recent work *git re-basin* (Ainsworth et al., 2022), where authors show that the loss landscape of neural networks contain (nearly) a single basin after accounting for all possible permutation symmetries of hidden

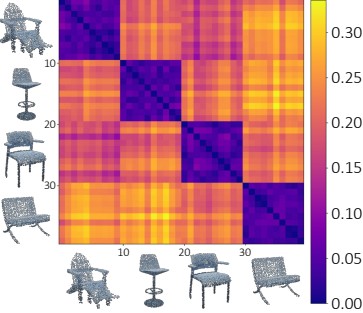

Figure 30: **L2 distances between `inr2vec` embeddings.** For each shape, we fit 10 INRs starting from the same weights initialization (40 INRs in total). Then we plot the L2 distances between the embeddings obtained by `inr2vec` for such INRs.

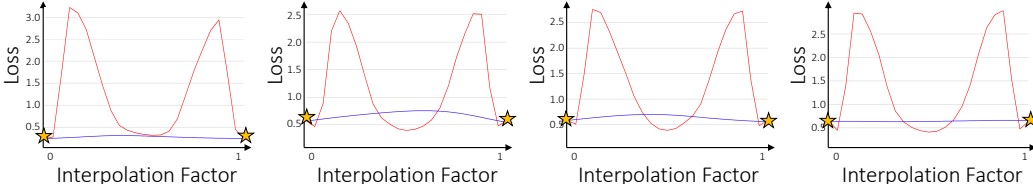

Figure 31: **Linear mode connectivity study.** Each plot shows variation of the loss function over the same batch of points when interpolating between two INRs representing the same shape. The red line describes the interpolation between INRs initialized differently, while the blue line shows the same interpolation between INRs initialized from the same random vector. The yellow stars represent the loss value of the boundary INRs.

units. The intuition behind this finding is that, given two neural networks that were trained with equivalent architectures but different random initializations, data orders, and potentially different hyperparameters or datasets, it is possible to find a permutation of the networks weights such that when linearly interpolating between their weights, all intermediate models enjoy performance similar to them – a phenomenon denoted as *linear mode connectivity*.

Intrigued by this finding, we conducted a study to assess whether initializing INRs with the same random vector, which we found to be key to inr2vec convergence, also leads to linear mode connectivity. Thus, given one shape, we fitted it with two different INRs and then we interpolated linearly their weights, observing at each interpolation step the loss value obtained by the *interpolated* INR on the same batch of points. For each shape, we repeated the experiment twice, once initializing the INRs with different random vectors and once initializing them with the same random vector.

The results of this experiment are reported for four different shapes in Fig. 31. It is possible to note that, as shown by the blue curves, when interpolating between INRs obtained from the same weights initialization, the loss value at each interpolation step is nearly identical to those of the boundary INRs. On the contrary, the red curves highlight how there is no linear mode connectivity at all between INRs obtained from different weights initializations.

(Ainsworth et al., 2022) propose also different algorithms to estimate the permutation needed to obtain linear mode connectivity between two networks. We applied the algorithm proposed in their paper in Section 3.2 (*Matching Weights*) to our INRs and observed the resulting permutations. Remarkably, when applied to INRs obtained from the same weights initialization, the retrieved permutations are identity matrices, both when the target INRs represent the same shape and when they represent different ones. The permutations obtained for INRs obtained from different initializations, instead, are far from being identity matrices.

All these results favor the hypothesis that our technique of initializing INRs with the same random vector leads to linear mode connectivity between different INRs. We believe that the possibility of performing meaningful linear interpolation between the weights occupying the same positions across different INRs can be interpreted by considering corresponding weights as carrying out the same role in terms of feature detection units, explaining why the inr2vec encoder succeeds in processing the weights of our INRs.

This intuition can be also combined with the finding of another recent work (Yüce et al., 2022), that shows how the expressive power of SIRENs is restricted to functions that can be expressed as a linear combination of certain harmonics of the first layer, which thus serves as basis for the range of learnable functions.

As stated above, the evidence of linear mode connectivity between INRs obtained from the same initialization leads us to believe that the weights of the first layer extract the same features across different INRs. Thus, we can think of using the same random initialization as a way to obtain the same basis of harmonics for all our INRs. We argue that this explains why it is possible to remove the first layer of the INRs presented in input to inr2vec (as empirically proved in Appendix I): if the basis is always the same, it is sufficient to process the layers from the second onwards, that represent the coefficients of the basis harmonics combination, as described in (Yüce et al., 2022).

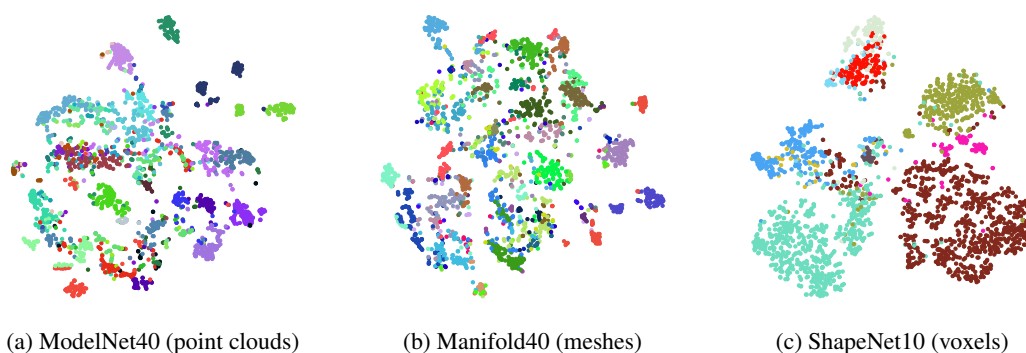

(a) ModelNet40 (point clouds)          (b) Manifold40 (meshes)          (c) ShapeNet10 (voxels)

Figure 32: **t-SNE visualizations of** `inr2vec` **latent spaces.** We plot the t-SNE components of the embeddings produced by `inr2vec` on the test sets of three datasets, ModelNet40 (left), Manifold40 (center) and Shapenet10 (right). Colors represent the different classes of the datasets.

## L   T-SNE VISUALIZATION OF `inr2vec` LATENT SPACE

We provide in Fig. 32 the t-SNE visualization of the embeddings produced by `inr2vec` when presented with the test set INRs of three different datasets. Fig. 32a shows this visualization for INRs representing the point clouds from ModelNet40, Fig. 32b for INRs representing meshes from Manifold 40, and Fig. 32c for INRs obtained from the voxelized shapes in ShapeNet10.

The supervision signal adopted during the training of our framework does not entail any kind of constraints *w.r.t.* the organization of the learned latent space. Indeed, this was not necessary for our ultimate goal – *i.e.*, performing downstream tasks on the produced embeddings. However, it is interesting to observe from the t-SNE plots that `inr2vec` favors spontaneously a semantic arrangement of the embeddings in the learned latent space, with INRs representing objects of the same category being mapped into close positions – as shown by the colors representing the different classes of the considered datasets.

## M   ABLATION ON INRS SIZE

Fig. 33 reports a study that we conducted to determine the size of the INRs adopted throughout our experiments. More specifically, we considered three alternatives of SIREN, all featuring 4 hidden layers but different number of hidden features, namely 128, 256 and 512 respectively.

In the figure we show how the three SIREN variants perform in terms of being able of representing properly the underlying signal, which in this example is the orange plane on the left. Since we needed to create datasets comprising a huge number of INRs, we considered both the quality of the representation as well as the number of steps of the fitting procedure, with the goal of finding the best trade-off between quality and fitting time.

The results presented in the figure highlight how a SIREN with 512 hidden features can learn to represent properly the input shape in just 600 steps, while the other variants either take longer (as in the case of 256 hidden features) or not obtain at all the same quality (when using 128 hidden features).

This experiment enabled us to draw the conclusion that a SIREN with 4 hidden layers and 512 hidden features is the proper tool to obtain a good quality INR in short time.

## N   SHAPE RETRIEVAL AND CLASSIFICATION ON DEEPSDF LATENT CODES

In Appendix A we show that frameworks that adopt a shared network to produce INRs struggle to obtain a good representation quality, while individual INRs do not suffer of this problem by design. In this section we goes one step further and consider the possibility of peforming downstream tasks on the latent codes produced by DeepSDF (Park et al., 2019).

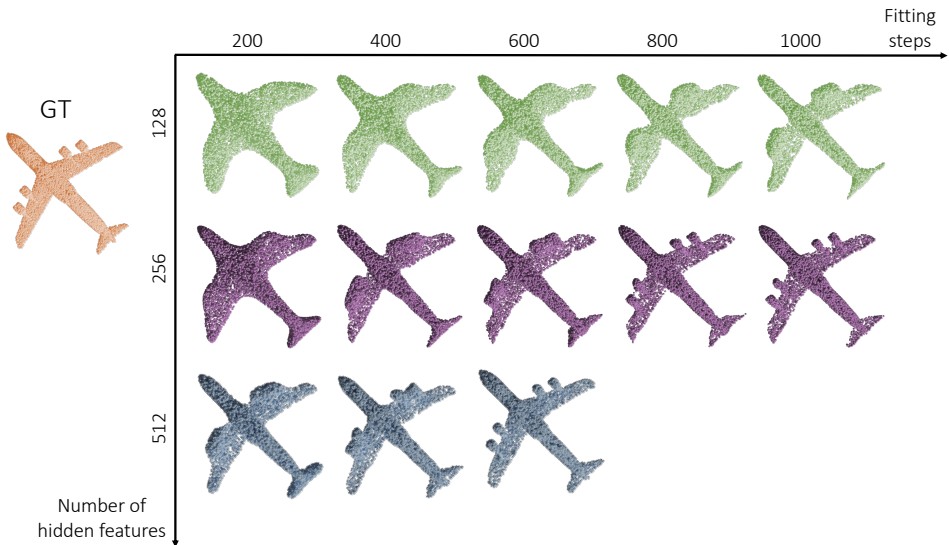

Figure 33: **Comparison between different hidden sizes for INRs.** We report the fitting steps needed to obtain a good representation for INRs featuring different number of hidden features.

| Method | ModelNet40 | | | | Method | ModelNet40 |
|---|---|---|---|---|---|---|
| | mAP@1 | mAP@5 | mAP@10 | | | Accuracy |
| PointNet (Qi et al., 2017a) | 80.1 | 91.7 | 94.4 | | PointNet (Qi et al., 2017a) | 88.8 |
| PointNet++ (Qi et al., 2017b) | 85.1 | 93.9 | 96.0 | | PointNet++ (Qi et al., 2017b) | 89.7 |
| DGCNN (Wang et al., 2019b) | 83.2 | 92.7 | 95.1 | | DGCNN (Wang et al., 2019b) | 89.9 |
| inr2vec | 81.7 | 92.6 | 95.1 | | inr2vec | 87.0 |
| DeepSDF | 69.8 | 85.4 | 89.8 | | DeepSDF | 64.9 |

Table 9: **Comparison between** inr2vec **and DeepSDF.** We report results in shape retrieval (left) and shape classification (right) when using standard baselines, inr2vec embeddings or DeepSDF latent codes.

In particular, we trained DeepSDF to fit the UDFs of our augmented version of ModelNet40, composed of ∼100K point clouds. For a fair comparison, we set the dimension of DeepSDF latent codes to 1024 – *i.e.*, the same used for inr2vec embeddings. Then we performed the experiments of shape retrieval and classification using DeepSDF latent codes, with the same settings presented in Section 4 for our framework.

The results reported in Table 9 highlight that the poor representation quality obtained with DeepSDF – and shown to be an intrinsic problem with shared network frameworks in Appendix A – has a detrimental effect on the quantitative results, proving once again that INR frameworks based on a shared network cannot be deployed as tool to obtain and process INRs effectively.

## O  SHAPE GENERATION: ADDITIONAL COMPARISON

In Fig. 8b we show a qualitative comparison between shapes generated with our framework (Section 4 - *Learning a mapping between* inr2vec *embedding spaces.*) and with competing methods, *i.e.*, SP-GAN (Li et al., 2021) for point clouds and OccupancyNetworks (Mescheder et al., 2019) for meshes.

In Fig. 34 we extend this comparison, by presenting samples obtained with our formulation applied to the voxelized chairs of ShapeNet10 and comparing them with samples produced by two additional methods that learn a manifold of individual INRs, namely GEM (Du et al., 2021) and GASP (Dupont et al., 2021b), for which we used the original source code released by the authors. To generate the figure, despite the sampled shapes being voxel grids, we adopt the same procedure used by GEM and GASP and reconstruct meshes by applying Marching Cubes to extract the 0.5 isosurface.

Fig. 34 show that all the considered methods can generate samples with a good variety in terms of geometry. However, it is possible to observe how the qualitative comparison favors the shape

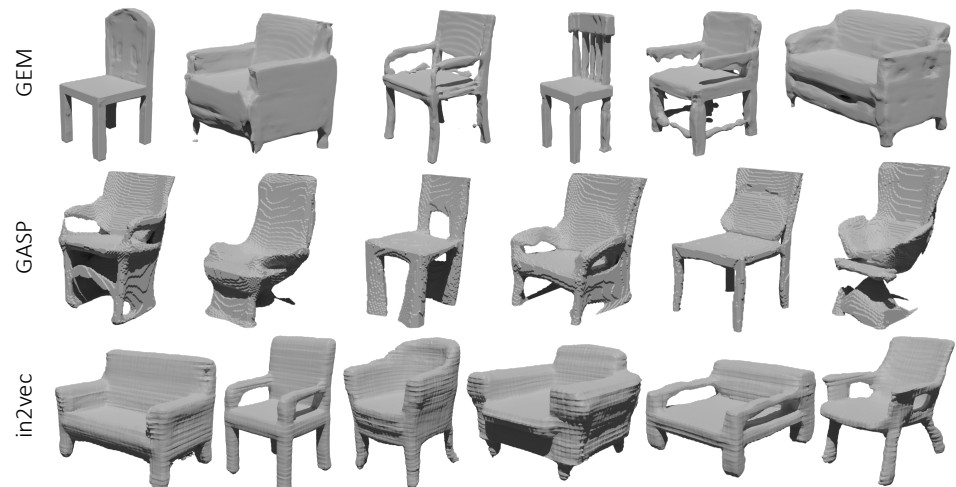

Figure 34: **Shape generation: qualitative comparison.** We show samples generated with GEM (Du et al., 2021), GASP (Dupont et al., 2021b) and with our method.

generated with `inr2vec`, that appear smoother than the ones generated by GASP and less noisy that the samples produced by GEM.

## P  INR CLASSIFICATION TIME: EXTENDED ANALYSIS

We report here the extended analysis of the inference times reported in Fig. 6, where we present the classification inference time needed to process *udf* INRs by standard point cloud baselines – PointNet (Qi et al., 2017a), PointNet++ (Qi et al., 2017b) and DGCNN (Wang et al., 2019b) – and by `inr2vec` encoder paired with the fully-connected network that we adopt to classify the embeddings (see Section 4).

The scenario that we had in mind while designing `inr2vec` is the one where INRs are the only medium to represent 3D shapes, with discrete point clouds not being available. Thus, in Fig. 6 for PointNet, PointNet++ and DGCNN we report the inference time including the time spent to reconstruct the discrete cloud from the input INR. In Fig. 35 and Table 10, for the sake of completeness, we report also the baselines inference times assuming the availability of discrete point clouds, stressing however that this is unlikely if INRs become a standalone format to represent 3D shapes.

The numbers plotted in Fig. 35 and reported in Table 10 show clearly that our framework presents a big advantage *w.r.t.* the competitors. Indeed, by processing directly INRs – where the resolution of the underlying signal is theoretically infinite – `inr2vec` can classify INRs representing point clouds with different number of points with a constant inference time of 0.001 seconds.

The considered baselines, instead, are negatively affected by the increasing resolution of the input point clouds. While the inference time of PointNet and PointNet++ is still affordable even when processing 64K points, DGCNN gets drastically slow already at 16K points. Furthermore, if point clouds need to be reconstructed from the input INRs, the resulting inference time become prohibitive for all the three baselines.

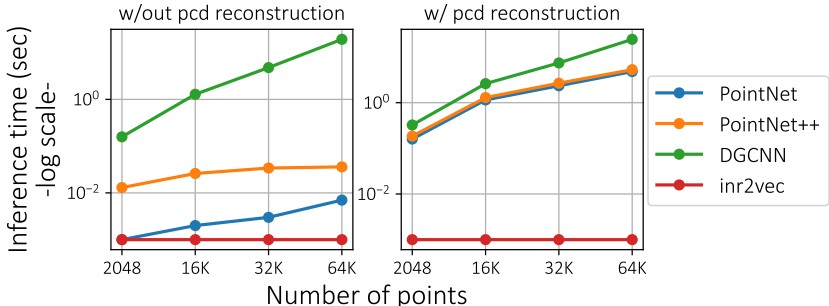

Figure 35: **Time required to classify INRs encoding udf.** We plot the inference time of standard baselines and of our method, both considering the case in which discrete point clouds are available (left) and the one where point clouds must be reconstructed from the input INRs (right).

| | Inference Time (seconds) | | | |
|---|---|---|---|---|
| Method | 2048 pts | 16K pts | 32K pts | 64K pts |
| PointNet | **0.001** | 0.002 | 0.003 | 0.007 |
| PointNet* | 0.171 | 1.315 | 2.609 | 5.230 |
| PointNet++ | 0.013 | 0.026 | 0.034 | 0.036 |
| PointNet++* | 0.185 | 1.293 | 2.672 | 5.287 |
| DGCNN | 0.158 | 1.285 | 4.788 | 19.26 |
| DGCNN* | 0.325 | 2.612 | 7.426 | 24.436 |
| inr2vec | **0.001** | **0.001** | **0.001** | **0.001** |

Table 10: **Time required to classify INRs encoding udf.** All the times are computed on a gpu NVidia RTX 2080 Ti. * indicates that the time to reconstruct the discrete point cloud from the INR is included.

