# OpenReview forum: "Deep Learning on Implicit Neural Representations of Shapes"
_ICLR.cc/2023/Conference — ICLR 2023 poster_

### Official Review · Reviewer_rRwy · 2022-10-14

**Confidence:** 4
**Correctness:** 4
**Technical Novelty And Significance:** 3
**Empirical Novelty And Significance:** 3
**Recommendation:** 8

**Clarity, Quality, Novelty And Reproducibility:**

**Clarity**

The paper is clearly written and the figures are very nice and informative.

**Quality**

The method is well explained, the experiments are thorough and well done and most relevant prior work is discussed. While there are a few weaknesses as listed in the previous sections, overall the paper is of high quality.

**Novelty**

While the idea of performing deep learning on INRs has already been considered in [1], the method presented here is quite different and interesting. Further, the authors consider a variety of different tasks not handled in [1] and also show more impressive empirical results on various 3D tasks.

**Reproducibility**

The appendix is very detailed and the paper seems to contain most details required to reproduce the results in the paper.

**Details Of Ethics Concerns:**

No ethical concerns.

**Strength And Weaknesses:**

**Strengths**
- With the growing interest in INRs, the paper tackles an important and timely problem (performing deep learning on INRs) which should be of interest to the ICLR community
- The paper is well written and clear
- The proposed method makes sense and is well suited for the tasks in the paper
- The experiments are well done and thorough. The authors tackle an impressively large number of tasks
- The appendix is very thorough and includes a lot of helpful extra clarifications as well as experimental results

**Weaknesses**
- Experimental results do not match SOTA methods on most tasks considered. However, this is explicitly mentioned as a weakness in the conclusion and I don't believe this is crucial. As this paper consists in one of the first attempts to perform deep learning on INRs, it is reasonable to not expect SOTA results
- As mentioned in the paper, the work is closely related to [1], which also proposes to perform various deep learning tasks on INRs. While it is briefly discussed in the paper, I believe readers would benefit from a more detailed comparison. For example, the authors mention that [1] rely on a shared network whereas the proposed method processes individual INRs. However, the encoder/decoder architecture is effectively shared across INRs too, and the encoding/decoding steps limit the reconstruction quality too. It would be nice to clarify this point in a little more detail in the paper. Further, while there are some experiments comparing the proposed model to [1] in the appendix, there are no comparisons on the tasks of classification and shape generation, both of which are tackled by [1]. For example, it seems like [1] also does classification on ShapeNet10 - why not include a comparison to this in the paper?
- The evaluation of the generative model is quite weak. There are no quantitative metrics and no qualitative comparisons to other baselines. While quantitative evaluation of generative models of 3D shapes is difficult, it would be nice to at least have qualitative comparisons. For example, [2, 3] follow a very similar setup to the paper (generating task/modality-agnostic INRs) and also have experiments on 3D shapes, which might be worth comparing with
- In the timing experiments (Figure 6), I think it makes sense to include the same experiments *without* the time required to reconstruct the point cloud from the INR. It is unclear from this experiment if most of the time is spent converting the INR to point clouds or if it's the actual point cloud model that is slow. While I understand the motivation for including the INR to point cloud conversion time, I think this experiment would greatly benefit from showing both the experiment with and without the conversion time

**Questions**
- As mentioned in Section 3, the encoder takes as input all the *hidden* layers of the INR and does not include the input and output layers. How are these layers treated? For example, given a latent embedding, you would still need to the input and output layer to reconstruct the signal. Are these just stored separately? I could not find any information explaining this in the paper, which I feel is a fairly important point

**Nitpicks**
- It seems like most citations are missing brackets. Using something like `\citep{}` should help
- Some of the wording can be slightly misleading. For example, in Section 4, "inr2vec is able to match, and in some cases slightly surpass, the considered baselines" does not really align with the results. It seems like the method underperforms PointNet++ (often by a significant margin) on 8/9 experiments. The wording should be changed to more accurately reflect this


[1] From data to functa: Your data point is a function and you can treat it like one

[2] Learning signal agnostic manifolds of neural fields

[3] Generative models as distributions of functions

**Summary Of The Paper:**

This paper proposes inr2vec, a framework for embedding implicit neural representations (INR) into a latent representation. The latent representations can then be used as data for various downstream tasks including classification, generative modeling and shape completion, effectively leading to a framework for performing deep learning directly on INRs. The authors learn such embeddings by training an autoencoder on the weights of INRs fitted to various 3D data, with the encoder mapping the weights to a latent vector and the decoder mapping the latent to the weights of an INR that is decoded into the data it represents. The authors perform a series of experiments on various 3D shape representations (point clouds, voxels, meshes) and various tasks (point cloud retrieval, classification, part segmentation, generative modeling, point cloud completion and surface reconstruction), with interesting results.

**Summary Of The Review:**

Overall I believe this is a strong paper which should be of interest to the ICLR community. The paper presents a novel method for performing deep learning on INRs with interesting experiments. While the results are not SOTA and there are a few other minor weaknesses, I still believe this paper is strong and as such I recommend acceptance.

---

> ### Author Response · Authors · 2022-11-15
> **Answer to Reviewer rRwy**
>
> We thank the reviewer for the valuable positive feedback and for acknowledging that inr2vec is one of the first attempts of performing deep learning on INRs, an important problem that should be of interest to the ICLR community. We address the weaknesses in the following.
>
> Weakness 1. We are glad the reviewer considered our attempts to perform deep learning on INRs valuable and does not see as critical that results are not SOTA. This is indeed the case also for the contemporary work addressing the same problem by Dupont et al. [1].
>
> Weakness 2. The reviewer pointed out correctly that the representation learning scheme used to train inr2vec relies on a shared network for the whole dataset, similar to approaches like [1] and others that we consider in Appendix A. The important difference between such approaches and our method, though, relies in the fact that, in shared network frameworks like DeepSDF [A], OccupancyNetworks [B], or [1], the shared network and the set of latent codes is the implicit representation, whose reconstruction quality is negatively affected by using a single network to represent the whole dataset (see Appendix A). In our framework, instead, we decouple the representations (INRs) from the possibility of processing them in downstream tasks (inr2vec). The quality of the representation is entrusted to the individual INRs and inr2vec does not influence it: as shown in the paper, the SIREN INRs that we use in our experiments already represent the underlying signal with good fidelity. Being individual INRs in charge of the representation, then inr2vec is only in charge of producing embeddings that are good to perform downstream tasks. As requested by the reviewer, we have discussed in more detail these analogies and differences with [1] in Appendix A. Finally, we argue that having decoupled representation and processing holds the potential for improving them independently.
>
> For what concerns a possible comparison on the ShapeNet classification experiment with the one reported in [1], while we agree on the relevance of such a study, we could not find a way to pursue it. The dataset adopted by the authors of [1] is an augmented version of the original ShapeNet, and the authors could not share it with us due to company policies (the authors of [1] were indeed very kind in answering quickly and precisely to our request). Moreover, the source code of [1] was not available at the time of the submission, it has been released recently – but the part to replicate the experiments on 3D shapes is not available.
>
> Weakness 3. As suggested by the reviewer, in Appendix O, we provide a qualitative comparison between ShapeNet chairs generated with our method and with [2] and [3], using the original code provided by the authors. The results highlight that all the methods are comparable for which concerns the variety of the generated geometries. However, the comparison shows that shapes generated with our method tend to be less noisy than the ones generated with [2] and with smoother surfaces than the ones generated with [3].
>
> Weakness 4. We reported in Appendix P an extended version of Fig. 6, including inference times without the point cloud reconstruction procedure. We agree that this extended analysis provides useful insights. Indeed, it is possible to observe that the methods that process the discrete point clouds (PointNet, PointNet++, and DGCNN) suffer from increasing the input resolution even without considering the time to reconstruct the point clouds from the input INRs. On the contrary, inr2vec yields a constant fast inference time.
>
> Question. As far as representation learning is concerned, the input and output layers are not needed (as shown in Appendix I). Moreover, thanks to reviewer VZbJ, we also found an interesting theoretical explanation for this, which is now reported in Appendix K. As far as signal reconstruction is concerned, our decoder can reconstruct the original signal only from the latent code as shown in Figure 4, 16, 17, 18, i.e., without the need to store the input and output layers. Indeed, this is exactly the learning objective used to train inr2vec.
>
> Nitpicks. We used the reviewer’s valuable suggestions to improve our paper.
>
> [A] Park et al. “Deepsdf: Learning continuous signed distance functions for shape representation.” CVPR 2019.
> [B] Mescheder et al. “Occupancy networks: Learning 3d reconstruction in function space.” CVPR  2019.

---

> > ### Comment · Reviewer_rRwy · 2022-11-20
> > **Thank you for your response**
> >
> > I thank the authors for their detailed response. I appreciate the changes made to the paper to address my concerns as well as the additional experiments. In particular, I think the additional figures in Appendix K (showing the reconstructions from the decoder) as well as the timing results in Appendix P help further clarify the strengths and limitations of the proposed method. Overall, I think the authors have done a great job at empirically verifying the strengths and weaknesses of their approach.
> >
> > I've also read all other reviews and the authors' responses to these. After reading these, I still believe this is a strong paper with interesting ideas and thorough experiments and maintain my recommendation for acceptance.

---

### Official Review · Reviewer_HTw1 · 2022-10-19

**Confidence:** 4
**Correctness:** 3
**Technical Novelty And Significance:** 3
**Empirical Novelty And Significance:** 2
**Recommendation:** 6

**Clarity, Quality, Novelty And Reproducibility:**

The paper is clearly written and easy to follow. The experiments are comprehensive except missing one important baseline. The technical novelty is limited but the paper tackles an interesting novel research problem.

**Strength And Weaknesses:**

Strengths:

- This paper tries to address an interesting problem --- how to process INRs directly for downsream tasks without reconstructing the discrete counterparts. Given that nowadays INRs are popular to represent 3D shapes, directly processing them is potential very useful, if can be done properly.

- The experiments are very comprehensive and lots of results are presented. The proposed framework is evaluated on many downstream tasks ranging from classification to generation.

Weaknesses:
- The proposed framework has some technical drawbacks that limits its application scope. For example, all input INRs need to be initialized with the same random vector (page 4 bottom). This requirement for training data is acceptable but it also applies to input INRs in testing time. This severely limits the generalization ability and such requirement is probably even not possible in real-world where INRs can come from anywhere. In addition, all input INRs need to have the same network structures since the encoder of inr2vec is a MLP of fixed structure.

- Though the experiments are very comprehensive and cover many tasks, the quantitative results are inferior to compared baselines on nearly all tasks. Then the proposed framework is less favourable. Traditional discrete representions are easy to obtain (e.g., point cloud scan) and are ready to be processed by existing models. INRs, on the contrary, are much hard to obtain as it requires network training but better represent the signal. So I would expect the proposed framework to have superior performance, such that it would be worth applying deep learning models directly on INRs.

- Missing one important baseline. Auto-decoder INRs, where a single network fits the whole dataset by condiitoning on different learned embeddings, is an important baseline. As the authors suggested in page 2 ("Earlier work in .... downstream tasks"), it is the natural solution to this research problem. Even though it may have clear drawbacks (i.e., insufficint capacity), it is a direct baseline that should be compared to, especially for those discriminative tasks.



**Summary Of The Paper:**

This paper proposes a framework that computes a compact latent vector for an input implicit neural representation for 3D shape (i.e., encoded with MLP weights). It's trained on a dataset of INRs via reconstruction loss, and is able to obtain a compact embedding from an unseen INR (but still from the same data distribution) at inference time. The produced embeddings can be readily used for downstream tasks like classification or segmentation without the need of decoding an actual discrete representation. Through experiments on several different tasks, the authors demonstrate that the proposed framework achieve results on-par with simple baselines using discrete representation.

**Summary Of The Review:**

This paper tries to address an interesting problem of directly applying deep learning models on INRs. The proposed framework is somewhat novel but has some clear limitations that may prevents its real-world applications. Experiments are comprehensive but the method gets inferior performance on nearly all tasks, casting doubts on whether we really need such framework in the first place. Overall, the paper's weaknesses outweighs its strengths. Therefore, I'm inclined to rejection.

Update: given that the authors' rebuttal addressed my major concerns, I have raised my score from 5 to 6.

---

> ### Author Response · Authors · 2022-11-15
> **Answer to Reviewer HTw1 (part 1/2)**
>
> We thank the reviewer for the careful review and the positive judgment on the experimental validation of our work. We are glad that solving downstream tasks on INRs without reconstructing the discrete signal was considered interesting by the reviewer. As for weaknesses, we provide detailed answers below.
>
> Weakness 1. The reviewer argues that the use of a fixed random vector at initialization limits the application scope of the proposed framework. We use a shared initialization as a simple but effective solution to align internal representations of INRs (as discussed in the response to reviewer VZbJ and in Appendix K). However, we do not see the use of a shared initialization as a practical limitation. As noted in the paper, there are already good practical reasons to use a shared, precomputed initialization, e.g., to drastically reduce training time by means of meta-learned initialization vectors, as done in [A] and in the contemporary work exploring the processing of INRs [B], or to obtain desirable geometric properties, as done in [C]. We also note that for CNNs the use of a shared set of weights is already a reality, as the majority of CNNs are pre-trained on Imagenet by downloading weights provided by deep learning frameworks: this shows how sharing a common initialization is technically easy once there is consensus on its usefulness. The reviewer also argues that the requirement of a shared initialization vector breaks one of the visions behind our work, i.e., that in the future we will be able to download a dataset of INRs from the internet on which we would like to run downstream tasks. We believe that, actually, our set of assumptions provides a practical scenario under which such a vision can become concrete. Images can be downloaded and processed everywhere thanks to standard formats. Standards dictate all the hyperparameters of the algorithms involved to store and decode them (e.g., in the case of JPEG, block size, use of Discrete Cosine Transform, number of frequencies to retain, etc). If INRs are to become a first-class representation to store and communicate 3D shapes similar standards may emerge. Such standards will likely define a set of common network architectures and training recipes, and they could seamlessly enforce a shared initialization, if deemed useful by the community to ease performing downstream tasks on INRs. Indeed, our paper already shows the feasibility of such a standard by fitting with high-fidelity shapes from 6 diverse datasets of three diverse discrete representations (point cloud, meshes, and voxels) by using exactly the same network architecture, training recipe, and initialization. Therefore, we are led to believe that the assumptions under which we demonstrated the feasibility of inr2vec are practically viable and could be adopted to realize the vision we foresee in the paper.
>
> Weakness 2. As also discussed in the answer to reviewer shEg, we feel the need to highlight that converting discrete representations to INRs and then performing downstream tasks on the INRs is not the intended use of inr2vec, since it would be a cumbersome and slow procedure, as correctly pointed out by the reviewer. The scenario that we had in mind while designing inr2vec is the one where INRs are a standalone representation stored and deployed for 3D shapes. In such a scenario, envisioned due to the well-known advantages of implicit representations (e.g., decoupling of memory requirements and signal resolution, the possibility of a unified representation for 3D shapes, the capability of representing surfaces whose topology is not known a priori), the community would share 3D shapes by simply communicating the weights of INRs and thus no discrete representation would be available. While of course, we cannot be sure that this scenario will become a reality, we consider valuable to explore whether and how INRs can be processed directly to perform downstream tasks without reconstructing the discrete representations from them. Such a procedure, in fact, would entail a cumbersome computational overhead as exemplified in Fig. 6 and Appendix P.

---

> > ### Author Response · Authors · 2022-11-15
> > **Answer to Reviewer HTw1 (part 2/2)**
> >
> > Weakness 3. In Appendix N we report the results of shape retrieval and classification performed on the latent codes produced by DeepSDF [D], i.e., the best-known realization of the baseline architecture proposed by the reviewer. More specifically, we trained DeepSDF to fit the UDFs of our augmented ModelNet40, which contains ~100K point clouds. Then we performed the tasks of retrieval and classification using the produced latent codes -- the ones for the train set were optimized during training and the ones for the test and validation sets were obtained with latent space optimization (LSO). We note that, compared to the fast inference of inr2vec, LSO is an expensive optimization procedure that required 12 hours to recover the test and validation latent codes. We conducted the two tasks following the same protocol adopted for inr2vec embeddings and described in Section 4. The results reported in Appendix N highlight how the insufficient capacity of a shared network framework like DeepSDF (that we discuss also in Appendix A) affects negatively the downstream tasks, with quantitative results significantly inferior to those obtained with inr2vec.
> >
> > [A] Sitzmann et al. “Metasdf: Meta-learning signed distance functions.” NeurIPS 2020.
> > [B] Dupont et al. “From data to functa: Your data point is a function and you can treat it like one.” ICML 2022.
> > [C] Gropp et al. “Implicit geometric regularization for learning shapes.” ICML 2020.
> > [D] Park et al. “Deepsdf: Learning continuous signed distance functions for shape representation.” CVPR 2019.

---

> > > ### Comment · Reviewer_HTw1 · 2022-11-21
> > > **Response to authors**
> > >
> > > I thank the authors for the detailed response. As the authors describe their vision more clearly, I'm now convinced that shared initialization and fixed network structure are not major limitations. The additional comparison to DeepSDF looks good. Given that my major concerns are addressed properly, I have raised my score.
> > >
> > > One minor concern I have is that how realistic the scenarios that the authors envision for the proposed technique to be useful. Even though INRs becomes a standalone representation, it has to come from somewhere. While meshes can be easily designed/edited by artists and point clouds can be obtained by scanning, I don't see how INRs can be easily obtained without accessing discrete data. Answering this question is probably out of scope of this paper, so I won't blame too much on this.

---

### Official Review · Reviewer_VZbJ · 2022-10-23

**Confidence:** 4
**Correctness:** 3
**Technical Novelty And Significance:** 3
**Empirical Novelty And Significance:** 4
**Recommendation:** 8

**Clarity, Quality, Novelty And Reproducibility:**

The paper is clear and well-written. The targeted problem setting and the proposed solution are novel and interesting. The architecture is simple for implementation. However, the training process may be cumbersome as each shape in the ShapeNet dataset needs to be trained to obtain the implicit neural representation.

**Strength And Weaknesses:**

Pros:
+ The paper targets an interesting yet not well-explored problem to directly encode the learned INR into a compact latent vector. The solution is simple yet effective. Thorough experiments are conducted to show empirically that the encoded feature captures the high-level cues inherent in the geometric neural representation.
+ The writing and organization are clear and straightforward. The figures well demonstrate the key components of the paper. Overall, I enjoy reading the paper.
+ The design of the encoder-decoder is interesting. Unlike traditional autoencoding that encodes the weight values in the embedding, the supervision for recovering the learned function may be the key to success. Besides, the transfer between different embedding spaces is also interesting and impressive.

Cons:
Though experimentally verified, the paper lacks explanations and theoretic proofs on several critical issues.
1. As proved in [A], each feature unit in the first layer serves as a single frequency basis, whose spectral support is gradually expanded after the non-linear activations. That is to say, the input layer is the most crucial part for an INR to define the spectral basis. It is hard to understand why the input layer should be discarded and why their use does not change the inr2vec performance (page 4). With slightly different input layers, the INRs define drastically different functions. Though the same initialization leads to linear mode connectivity, I don't think initialization with the same random vector can mitigate this issue.
2. The property of the embedding space is not clearly demonstrated or analyzed (e.g., through t-sne). The invariance against noise, transformation, or the over-parameterized network (a shape can be represented by multiple network parameters) is unknown. According to the experimental results, the embedding captures both local (part segmentation) and global (classification and retrieval) information. The distance within the embedding space reflects the geometric/categorical similarity (interpolation and retrieval). However, it is unclear how such an ideal embedding space is guaranteed by the proposed supervision manner.


Minor issues:
1. SIREN with 256 units in each layer is able to recover nice geometric details in scene scale. I don't understand why the paper adopts a (512 units with 4 hidden layers) network configuration (page 6) for the coarse ShapeNet geometry.
2. INRs provide fine-grained and continuous geometric cues. However, the learned embeddings perform worse than the compared point cloud features. Though I think the novelty is sufficient, remarks concerning this issue and insights for future potentials should be added.

[A] A structured dictionary perspective on implicit neural representations. Gizem Yüce*, Guillermo Ortiz-Jimenez*, Beril Besbinar and Pascal Frossard, CVPR 2022

**Summary Of The Paper:**

The paper targets a novel {inr2vec} problem setting: to squeeze the weights of a learned implicit neural representation (INR) into compact latent codes for various downstream tasks. By training an encoder-decoder to recover the learned implicit function represented by the INR, the encoded feature can be fed into learning pipelines to solve tasks such as classification, retrieval, part segmentation, unconditioned generation, surface reconstruction, and completion. Detailed experiments verify the efficacy of the proposed method.

**Summary Of The Review:**

Overall, I think it is an interesting paper and is worth spreading in the community. The main concerns are summarized in the [Cons] section. I am currently positive, and I would like to see how the author responds and the opinions of other reviewers.

---

> ### Author Response · Authors · 2022-11-15
> **Answer to Reviewer VZbJ**
>
> We thank the reviewer for the positive judgment of our work and the insightful comments. We provide additional explanations and link to the theoretical results highlighted by the reviewer.
>
> Cons 1. Thanks for the reference to [A], we were not aware of it. It is indeed a very interesting characterization of INRs that applies also to the SIREN formulation we deploy. We also agree with the intuition that the same random initialization enables linear mode connectivity. Indeed, we are currently studying this phenomenon in a follow-up project, and we can share here preliminary results.
>
> If we fit an INR to the same shape multiple times starting from the same random vector, we indeed observe linear mode connectivity, as shown in Appendix K. Moreover, if we compute the alignment permutations that lead to linear mode connectivity as recently proposed in Git Re-Basin [B], in particular by the matching weight method, we find that no permutation is needed to align INRs of the same shape obtained from different trainings starting from the same init (the estimated permutation matrices are all identities). This implies that the first layer of our SIRENs and, consequently, also the resulting integer harmonics, which define the spectral basis on the function represented by the INRs, are consistently aligned across diverse fittings of the same shape. Moreover, we obtain the same alignment (all permutations are identities) even across diverse shapes when using the same initialization: a remarkable result on its own, that we plan to explore in more depth. Conversely, the permutation matrices are actual permutations (i.e., not identities) when the same initialization is not used. We believe that these findings provide an elegant explanation as to why in2vec performance is insensitive to the presence of the input layer when using the same initialization: since the features computed by the first layer are aligned across the dataset, the spectral basis is the same, and therefore all the information to compute a meaningful embedding is in the combining weights, i.e., in the hidden layers.
>
> Cons 2. As noted by the reviewer, we did not impose any kind of constraints on the organization of the latent space learned by inr2vec. Indeed, this was not necessary for the main purpose of our work, i.e., compressing INRs into encodings whose smaller size makes them amenable to be used in deep learning pipelines. Indeed, the only property we enforce through the loss is distinctiveness at instance-level. However, the representation learning scheme that we adopted to train inr2vec turns out to produce a properly structured latent space that enables smooth interpolations and can be used to perform a multitude of non-trivial task e.g., shape generation or meshing via learning a mapping between different inr2vec spaces. Similar properties have been observed when applying basic representation learning approaches to 3D data encoded as point clouds, without any need for additional losses [C]. As suggested by the reviewer, we report in Appendix L the t-SNE plots of inr2vec latent spaces for ModelNet40, Manifold40, and ShapeNet10. The clustered organization of the embeddings shown in the plots confirms the emergence of a reasonably disentangled class structure without any loss enforcing it explicitly.
>
> For what concerns the invariance properties of inr2vec, the results presented in Fig. 30 and described in Appendix K already provide empirical proof that our encoder is invariant to INRs obtained with different fitting runs, under the assumption of using the same weights initialization. We believe that a more general invariance to noise and transformations is indeed an interesting direction, but we consider it orthogonal to our work and we hope to address it in future work.
>
> Minor 1. The supplementary material of the SIREN paper shows how different SIREN sizes are needed for different target shapes (e.g., 5 layers with 256 hidden features for the thai statue and 5 layers with 1024 hidden features for the room). In Appendix M, we report the study that we conducted to select the size of SIREN adopted throughout our experiments. The reported results highlight how, even if a 4-layer SIREN with 256 hidden units is indeed already capable of representing fine-grained details, using 512 hidden features leads to faster convergence. Since we had to create many datasets composed of a huge number of INRs, we selected 512 as our hidden size, to exploit the faster convergence in order to speed up the dataset creation process.
>
> Minor 2. We already reported the inferior performance as a limitation in the Concluding Remarks. We slightly reworded it to highlight that this happens despite the ability of INRs to capture fine-grained and continuous cues.
>
> [B] Ainsworth et al. “Git Re-Basin: Merging Models modulo Permutation Symmetries”, https://arxiv.org/abs/2209.04836.
> [C] Achlioptas et al. “Learning representations and generative models for 3d point clouds.” ICML 2018.

---

> > ### Comment · Reviewer_VZbJ · 2022-12-01
> > **Thanks for the detailed response**
> >
> > I thank the authors for the detailed response. I appreciate the thorough experiments added in the Appendix and the patient explanations that well address all concerns raised by the reviewers. With these additional results, I think the paper empirically verifies its efficacy and delineates multiple interesting future potentials for in-depth exploration. I've learned a lot from the paper, and I believe it is a strong paper and worth spreading in the community.

---

### Official Review · Reviewer_shEg · 2022-10-24

**Confidence:** 4
**Correctness:** 3
**Technical Novelty And Significance:** 3
**Empirical Novelty And Significance:** 3
**Recommendation:** 6

**Clarity, Quality, Novelty And Reproducibility:**

In total, the writing and organization is good. The raised problem is also interesting. Experiments also show very interesting results.

**Strength And Weaknesses:**

Strenth:
* The trial to encode INRs weights to a compact code and try to use it for downstream tasks is interesting.
* The experiments well-verifies the possibility.

Weaknesses:
* If this is a right direction, then I have no concerns. My major concern is: it is not clear why we need such a trial.  Point cloud is the raw 3D data format which can be directly captured by 3D sensors. And, mesh is the raw data format created from some 3D modeling tools, such as Maya and 3DMax. Then, if we can do classification on point cloud and mesh, why we need to convert them into INRs and then conduct downstream tasks. If the conversion between different representations become easier, then it is also no need to propose a unified representation.
* The results do not show better performance.

**Summary Of The Paper:**

As implicit neural representations (INRs) is now becoming a popular 3D representation, especially for reconstruction tasks. This paper presents an interesting question: is it possible to use INR to replace explicit representation (like point cloud and mesh) for downstream tasks, such as classifaction or segmentation? Thus, a trial is presented: a method is designed for encoding INR weights into a vector which can be used for several tasks.

**Summary Of The Review:**

In whole, I appreciate the efforts for this trial. It is indeed an interesting attempt. However, I have only one concern is the necessity to do downstream tasks using INRs.

---

> ### Author Response · Authors · 2022-11-15
> **Answer to Reviewer shEg**
>
> We thank the reviewer for the overall positive judgment of our work. In this response, we justify why we believe it is important to research whether and how downstream tasks can be solved on INRs.
>
> Weakness 1. The reviewer asks why we need to convert a point cloud or a mesh into an INR to classify it or solve other downstream tasks. We agree that there are no practical advantages in following this path to perform shape classification or solve other downstream tasks, and we stress that this is not the intended usage of our framework. The scenario we envisage deals with raw discrete representations not being available alongside the corresponding INR. In other words, the scenario for which we propose our framework is one where INRs will be used as a standalone format to store and communicate 3D shapes, i.e., only the weights of the network will be available. We point out that a similar scenario has been considered in [A] and [B], where the authors focus on investigating how to process directly INRs without availability of the corresponding explicit representations.
>
> In this scenario, given a set of INRs shipped without the raw discrete data from which they originated, there are only two possible paths to perform downstream tasks: either convert the INR back into an explicit discrete representation and use deep learning models trained for it, e.g., extract the point cloud from the INR and then use a PointNet classifier, or be able to run deep learning models directly on INRs, as we do in our inr2vec framework. Besides being an interesting research question on its own, as agreed also by the reviewer, the second option has the practical advantage of being extremely faster to run, as shown e.g., by Figure 6 (detailed in Appendix P), where classification on INRs runs at least two orders of magnitude faster than extracting the clouds and classifying them, while delivering classification performance that is only 3-4 points of accuracy worse on average. Hence, we believe our method already delivers a very good trade-off between accuracy and running time in the considered scenario. Of course, it will be ideal to have a faster method that does not incur a performance penalty, and we hope that our study will call for other researchers to improve over our result for this novel and challenging problem. At the same time, we hope the reviewer will consider our paper a valuable step to achieve this end goal.
>
> We believe the scenario where INRs will be an independent medium to store and communicate 3D shapes is likely in the near future due to INRs, in particular individual INRs, being very effective at representing the 3D surface, while enjoying several benefits over discrete representations, which motivates the overwhelming research enthusiasm around them: they decouple the memory cost of the representation from the actual spatial resolution, thereby, e.g., supporting seamlessly multiple level-of-detail while rendering [C],  they have the capability of representing surfaces whose topology is not known a priori and the same neural network architecture can be used to fit different implicit functions, holding the potential to provide a unified framework to represent 3D shapes. We note here that today it is not easy to convert between representations: in particular, there are several meshing algorithms to go from point clouds to meshes, but they are usually slow and/or sensitive to the choice of hyperparameters and each of them has its own pros and cons.
>
> Weakness 2. We agree with the reviewer, as pointed out in the Concluding Remarks of our paper. We also note that contemporary work [A] addresses similar problems, e.g., classification of INR encoding voxel grids, which shows how the problem has already been considered interesting and relevant by the community, despite the results not being state of the art against established baselines for specific discrete modalities.
>
> [A] Dupont et al. “From data to functa: Your data point is a function and you can treat it like one.” ICML 2022.
> [B] Yang et al. “Geometry processing with neural fields.” NeurIPS 2021.
> [C] Takikawa et al. “Neural geometric level of detail: Real-time rendering with implicit 3D shapes.” CVPR 2021.

---

> > ### Comment · Reviewer_shEg · 2022-12-01
> > **Thanks for the detailed response**
> >
> > I appreciate the detailed response of my concerns, especially for weakness 1. I am not sure if INRs will become the standlone format in the future. But if yes, I agree this paper proposes an interesting idea. After reading others' comments, I am fine with the acceptance.

---

### Decision · Program_Chairs · 2023-01-20

**Decision:**

Accept: poster

**Justification For Why Not Higher Score:**

The paper presents an interesting idea. However, there are still some concerns on how practical the approach can actually be.

**Justification For Why Not Lower Score:**

N/A

**Metareview: Summary, Strengths And Weaknesses:**

The paper presents to use INR to replace explicit 3D representation for general downstream tasks. The reviewers generally agrees the paper presents an interesting idea and the experiments are conducted comprehensively. There remains some concerns on how practical the approach will be in terms of real-world application. The reviewers generally agree to accept the paper and the AC agrees.

**Note From Pc:**

if the above contains the word "oral" or "spotlight" please see: "oral" presentation means -> notable-top-5% and "spotlight" means -> notable-top-25%. As stated in our emails, we are disassociating presentation type from AC recommendations